evolution, health and disease and epidemiology

trade-offs, virulence, transmission, emerging zoonotic disease, evolution

**Author for correspondence:**
Elisa Visher
e-mail: elisa_visher@berkeley.edu

# The three Ts of virulence evolution during zoonotic emergence

Elisa Visher[1], Claire Evensen[4], Sarah Guth[1], Edith Lai[2], Marina Norfolk[3], Carly Rozins[5], Nina A. Sokolov[1], Melissa Sui[3] and Michael Boots[1,6]

[1]Department of Integrative Biology, [2]College of Natural Resources, and [3]College of Letters and Sciences, University of California, Berkeley, CA 94720, USA
[4]Mathematical Institute, University of Oxford, Oxford OX2 6GG, UK
[5]Department of Science and Technology Studies, Division of Natural Science, York University, Toronto, Ontario, Canada M3J 1P3
[6]Centre for Ecology and Conservation, College of Life and Environmental Sciences, University of Exeter, Penryn Campus, Penryn TR10 9FE, UK

  EV, 0000-0003-3984-4748; CE, 0000-0002-2060-2362; SG, 0000-0001-5533-9456; EL, 0000-0002-0968-4547; MN, 0000-0001-6034-3417; CR, 0000-0003-1503-4871; NAS, 0000-0002-2920-3106; MS, 0000-0002-3422-2723; MB, 0000-0003-3763-6136

There is increasing interest in the role that evolution may play in current and future pandemics, but there is often also considerable confusion about the actual evolutionary predictions. This may be, in part, due to a historical separation of evolutionary and medical fields, but there is a large, somewhat nuanced body of evidence-supported theory on the evolution of infectious disease. In this review, we synthesize this evolutionary theory in order to provide a framework for clearer understanding of the key principles. Specifically, we discuss the selection acting on zoonotic pathogens' transmission rates and virulence at spillover and during emergence. We explain how the direction and strength of selection during epidemics of emerging zoonotic disease can be understood by a three Ts framework: trade-offs, transmission, and time scales. Virulence and transmission rate may trade-off, but transmission rate is likely to be favoured by selection early in emergence, particularly if maladapted zoonotic pathogens have 'no-cost' transmission rate improving mutations available to them. Additionally, the optimal virulence and transmission rates can shift with the time scale of the epidemic. Predicting pathogen evolution, therefore, depends on understanding both the trade-offs of transmission-improving mutations and the time scales of selection.

## 1. Introduction

Throughout the current global pandemic of SARS-CoV-2, we have seen a growing public fascination with the role of pathogen evolution during disease emergence. In May 2020, reports of a mutational variant (D614G) increasing in frequency sparked concern about virus evolution [1] and more potentially adaptive variants have since been reported [2]. These experiences with SARS-CoV-2 and with previous epidemics of other zoonotic diseases have clearly demonstrated the potential for pathogens to evolve during disease emergence [3]. Despite this importance, public conversations around pathogen evolution are often fraught with misunderstandings. To some extent, this is likely reflective of the historical separation of evolutionary and medical disciplines [4]. Beyond that, however, scientific communication around pathogen evolution is particularly tricky because the science to be communicated provides no clear answers to be packaged into simple explanations.

Experts studying infectious disease evolution understand that pathogens have the potential to rapidly adapt due to high population sizes, short generation times, and relatively high mutation rates [5] and recognize that human

populations impose novel, although often understood, selection pressures [6]. At the same time, however, many experts are sometimes quick to express scepticism when public conversation is dominated by concern over pathogen evolution. This is partially because pathogen evolution is just one factor of many that collectively influence epidemic progression, so communication around its importance sits on a teeter-totter of balancing concern and attentiveness against a blinded focus on potential evolution over other factors shaping the epidemic [7,8].

Additionally, many experts studying infectious disease evolution are often quick to emphasize that we cannot predict how a specific pathogen will evolve [9]. This, however, does not mean that we have absolutely no idea of how pathogens generally may evolve. We expect that pathogens will evolve in response to selection in human populations, but the speed at which they do depends critically on the availability of adaptive variation and the relative strength of selection compared to stochasticity, both of which relate to the number of infected individuals [10]. Theory predicts that pathogens may evolve towards optimal virulence and transmission rates due to underlying constraints, but these predictions depend on nuances of pathogen biology, epidemic stage, and host population structure [11,12]. It can, understandably, be frustrating when asking how a pathogen will evolve to hear predictions that sound like contradictions and non-answers, but this reflects the complicated realities of pathogen evolution. However, this real uncertainty also seems to have created an environment where hope for simple answers means that misinformation can spread.

On top of the inherent challenges of communicating complex scientific concepts, researchers studying pathogen evolution must also play 'whack-a-mole' against a variety of misconceptions that are wrong in different ways. Public concern sometimes skews towards pathogens evolving to be hyper-virulent, hyper-transmissible superbugs [13]. Alternatively, historical theories of evolution towards avirulence still pervade the public consciousness and sometimes lead to the prediction that pathogens universally evolve to become less dangerous [14]. In both directions, these misconceptions can lead to inappropriate public health policies. However, the disjointed nature of combatting misconceptions as they arise has led to much of the conversation on pathogen evolution in emerging zoonotic diseases being scattered across the scientific literature and media. This can be compounded by the fact that researchers studying pathogen evolution come from a variety of sub-disciplines and their work is often not well integrated [15].

As pathogen evolution continues to be an important conversation in the current pandemic of SARS-CoV-2 and is likely to again be important during future epidemics of emerging zoonotic disease, this review aims to collect insights from the wealth of research on pathogen evolution to provide a centralizing, conceptual understanding of the factors shaping the evolution of transmission rate and virulence in epidemics of novel zoonotic disease. While we cannot comprehensively discuss this vast literature, our aim is to provide a framework so that readers understand the general principles of pathogen virulence and transmission evolution and can also see how variations in the assumptions of these models based upon nuances of biology and population structure can lead to deviations in their predictions. Because

strong reviews of virulence evolution exist elsewhere in the literature [4,12], our review focuses specifically on virulence evolution in epidemics of novel zoonotic disease to focus on how general theory for virulence evolution is altered by the specific characteristics of emerging zoonotic diseases and shifting selection pressures during epidemics. Extending beyond the scope of any single theoretical paper on this topic, we will discuss: (i) how do trade-offs between pathogen traits constrain pathogen evolution? (ii) What predicts pathogen virulence at the spillover barrier? (iii) Why it is hard to predict how novel zoonotic pathogens will evolve? And (iv) how do optimal strategies in populations with different epidemiological characteristics change over time during an epidemic? Through this, we describe predictions for pathogen evolution during epidemics of emerging zoonotic disease and how they can change depending on pathogen biology and host population structure.

## 2. The three Ts framework: trade-offs, transmission, and time scales

The adaptive evolution of any trait depends on the presence of variation and the ability of selection to act on that variation. It is clear that pathogens, particularly RNA viruses, can quickly generate and maintain large amounts of variation [16]. At the start of an epidemic, selection on these variants is weak compared to stochastic and demographic pressures, but gains strength as the number of infections increase [10]. Selection on virulence during epidemics of emerging zoonotic disease can be understood by considering the 'three Ts': trade-offs, transmission, and time scales [7,17–19]. See figure 1 for a graphical summary.

In terms of *trade-offs*, theory has often assumed, and empirical data have increasingly shown us, that many pathogen traits, like transmission rate and virulence, trade-off with each other [12,17,20,21] (table 1). The trade-off theory is important because it explains how different intermediate virulence, transmission, and recovery rates can be optimal for a pathogen due to constraints between these key traits [12,17,21]. In terms of transmission, emerging zoonotic pathogens typically do not have histories of selection in human populations and thus are likely to be maladapted for human-to-human transmission [40]. This maladaptation potentially means that emerging zoonotic pathogens may initially have 'no-cost' mutations available that improve transmission rate without impacting traits like virulence [18]. In these cases, emerging diseases can be selected to increase their *transmission rates* with no, or potentially counterintuitive, impacts on virulence [18]. Finally, *time scale* matters since, even with trade-offs between virulence and transmission rate, transmission rate improvements continue to be the most important selection pressure at the start of an epidemic because the relative strength of selection on transmission rate and virulence shifts as the density of susceptible hosts changes during an epidemic [19,41]. This effect further alters a number of theoretical predictions that are classically evaluated at equilibrium for how different host, pathogen, and epidemiological factors shape selection on pathogen traits. Therefore, a pathogen's optimum strategy changes over *time* during an epidemic under a wide array of conditions. We will discuss each of these in detail below.

**Figure 1.** The three Ts of virulence evolution during zoonotic emergence. Trade-offs between virulence and transmission rate determine pathogen fitness at every point during an epidemic, regulating pathogen fitness at the spillover barrier and shaping selection as the epidemic progresses. Early in the epidemic, however, individual transmission rate improving mutations may be 'costless' and not have trade-offs. Improvements in transmission rate are the most important selection pressure during epidemic take-off and building phases, though selection is weak at take-off. Finally, the time scale of the epidemic shifts the pathogen's optimal virulence and transmission rate strategies as the density of susceptible hosts changes. Created with Biorender.com. (Online version in colour.)

## 3. How do trade-offs between pathogen traits constrain pathogen evolution?

Evolutionary biologists have long been interested in why pathogens harm their hosts or cause virulence (figure 2) [42]. Based on the assumption that host damage was detrimental to parasite fitness, early ideas predicted that all parasites should evolve towards avirulence [4,14]. This was considered the 'conventional wisdom' until the 1980s, when foundational papers began to appreciate that virulence might be linked to other parasite traits like transmission or recovery rates and, therefore, could have an evolutionary optimum [17]. Trade-offs between these traits would mean that low virulence would come at a cost of low transmission rate or fast recovery and that avirulence would, therefore, hinder parasite fitness. This virulence and transmission trade-off is now fundamental to our theories on pathogen evolution.

Theory on the virulence and transmission trade-off typically suggests that virulence and transmission rate are both functions of the within-host exploitation or replication rate [4,12]. Because faster replicating pathogens generate larger population sizes, they increase their transmission rate while causing more host damage [12,21]. Damage increases host mortality, thereby decreasing the host's infectious period and providing a shorter window for the infected host to contact susceptible hosts [17]. In short, faster within-host

replication increases the likelihood of infection upon contact while decreasing the overall duration of infection [17,21]. Under the trade-off hypothesis, parasites are therefore selected for exploitation rates that balance virulence and transmission rate [12,17,21].

Transmission rate and virulence do not necessarily need to trade-off through the within-host exploitation rate for selection to balance the two traits. A virulence–recovery trade-off can occur if low replication rates make pathogens easier to clear such that lower virulence trades off with faster recovery rates [17]. Alternatively, a transmission–recovery trade-off can occur if the immune response is activated in a density-dependent manner so that high replication rates have high transmission rates, but fast recovery [43]. A sickness behaviour-transmission trade-off may result if faster replication rates make the host feel sick and isolate themselves so that high replication leads to a higher probability of infection upon contact, but fewer contacts [44]. Finally, the virulence and transmission trade-off does not necessarily depend on changes to the within-host replication rate if symptoms themselves are needed for transmission [28].

In simple host–parasite models, pathogens are selected to maximize the epidemiological $R_0$ (i.e. the number of secondary infections that a parasite produces during its infectious period in an entirely susceptible population) [17] (but see [45,46]). The virulence–transmission trade-off predicts that these two traits are positively correlated, but the shape of this relationship is critical to the predictions of evolutionary theory [17,21]. When the trade-off is linear, pathogens evolve maximum virulence; but when the trade-off is saturating (such that virulence is acceleratingly costly in terms of transmission rate), pathogens will evolve towards an intermediate virulence [4,17]. Given the centrality of the trade-off hypothesis to our understanding of virulence, it is noticeable that there are a number of empirical studies that have found support for the core idea (table 1, rows 1–2) [20].

## 4. What predicts virulence and transmission rate at spillover?

### (a) Virulence and transmission trade-offs act at spillover

As we have outlined, theory on the virulence and transmission trade-off is based upon the idea that pathogens will be selected towards an optimal level of virulence within the host populations to which they are adapted [12]. Recently emerged zoonotic diseases do not have this evolutionary history with human populations and are, therefore, highly unlikely to be at their evolutionary optimum when they first emerge [40,47]. However, emerging pathogens may still be regulated by an underlying virulence and transmission trade-off. In meta-analyses of recently emerged viral zoonoses, excessively high virulence is associated with a lower $R_0$ [40,48,49] and this negative association supports the theoretical prediction that high virulence impedes pathogen fitness. Theory also predicts a cost to excessively low virulence, an effect that is not supported in these analyses [17,40]. However, this could easily result from discovery bias because we are unlikely to notice low $R_0$ zoonoses that cause only a few infections and have low virulence [11]. As such, there is little evidence to not expect emerging diseases to be governed by trade-offs once they emerge into human populations.

**Table 1.** Empirical tests of virulence evolution theory.

| key finding | key empirical evidence (selected papers) |
| --- | --- |
| virulence and transmission rate are positively correlated through replication rate | *Mus musculus/Plasmodium chabaudi* [22]; *Homo sapiens/Plasmodium falciparum* [23]; *Daphnia magna/Pasteuria ramosa* [24]; *Homo sapiens/*HIV-1 [25]; *Danaus plexippus/Ophryocystis elektroscirrha* [26]; meta-analysis of multiple systems [20] |
| positive trait correlations saturate so that $R_0$ peaks at intermediate virulence | *Oryctolagus cuniculus/*Myxoma virus [17] (virulence–recovery rate); *Homo sapiens/Plasmodium falciparum* [23] (virulence–transmission rate); *Daphnia magna/Pasteuria ramosa* [24] (virulence rate–transmission rate); *Homo sapiens/*HIV-1 [25] (virulence rate–transmission rate), *Danaus plexippus/Ophryocystis elektroscirrha* [26] (virulence–transmission rate), *Gallus gallus domesticus/*Marek's disease virus [27] (virulence–transmission rate), *Haemorhous mexicanu/ Mycoplasma gallisepticum* [28] (virulence–transmission rate) |
| high susceptible density at the start of an epidemic selects for higher virulence | *Escherichia coli/*bacteriophage lambda [29] |
| structured host populations select for less transmissible, prudent strategies | *Escherichia coli/*T4 coliphage [30]; *Plodia interpunctella/*granulosis virus [31]; *Escherichia coli/* bacteriophage lambda [32] |
| high virulence can trade-off with decreased host movement | *Danaus plexippus/Ophryocystis elektroscirrha* [33]; *Haemorhous mexicanu/Mycoplasma gallisepticum* [34]; *Paramecium caudatum/Holospora undulata* [35] |
| virulence evolves in natural epidemics of emerging disease | *Haemorhous mexicanu/Mycoplasma gallisepticum* [36,37] (less virulent strains spread fastest because of movement–virulence trade-offs and then are replaced by higher virulence strains. When hosts start evolving resistance, virulence continues to increase through increased symptom severity rather than through replication rate) *Oryctolagus cuniculus/*Myxoma virus [38] (lower virulence quickly evolves from extremely high virulence introduction strains. When hosts start evolving resistance, virulence starts to increase) *Corvus brachyrhynchos/*West Nile Virus [39] (a mutation conferring high virulence in American crows was positively selected, though this may have been a result of selection in another bird or vector species) |

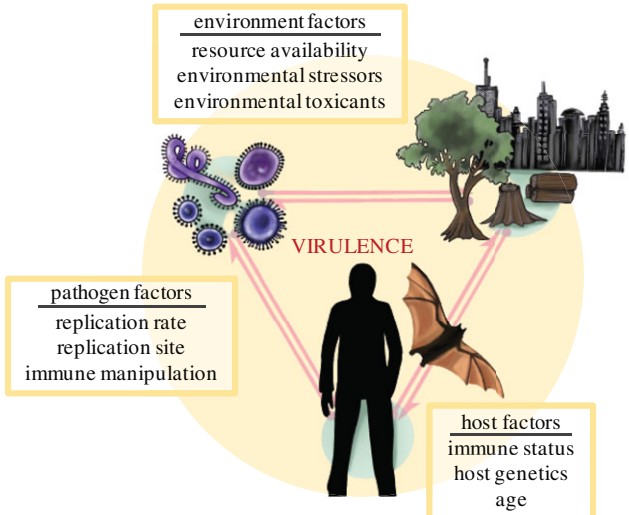

**Figure 2.** Disease triangle of virulence. (Online version in colour.)

## (b) Virulence and transmission rates of zoonotic pathogens reflect evolutionary histories with their reservoir hosts

Emerging zoonoses vary widely in their virulence and transmission rates, but there are key reservoir host characteristics that are associated with the pathogen's phenotype in humans [40,48,50]. In particular, meta-analyses of recently emerged viral zoonoses have supported phylogenetic trends in zoonotic potential [40]. The phylogenetic distance between a pathogen's reservoir host and novel host predicts the pathogen's probability of being zoonotic [50], virulence [40,51], and $R_0$ [40,48]. Mammalian hosts closely related to humans (e.g. primates) harbour zoonoses associated with lower human mortality and higher $R_0$, while more distantly related hosts (most notably, bats) harbour highly virulent zoonoses that appear to be relatively maladapted for human-to-human transmission [40,52]. These phylogenetic trends can be understood if pathogens from distantly related reservoir hosts have evolved replication strategies adapted to their reservoir host's more dissimilar immunology, physiology, and ecology [40,47].

Importantly, these variations in pathogen virulence upon emergence reflect evolutionary histories within non-human reservoir hosts and demonstrate that emerging zoonotic diseases are not likely to be well adapted to human populations [40,47]. Reservoir host and pathogen traits can suggest what phenotypes a pathogen may have upon emergence, but do not tell us where these starting point phenotypes are relative to a pathogen's 'ideal' phenotypes in humans, since each pathogen will have a different evolutionary optimum depending on the nuances of its biology in the new host [9]. Because we cannot know where an emerging pathogen's starting point phenotypes are relative to its

optimal phenotypes, we cannot precisely predict the direction of selection on virulence or transmission rate.

## 5. Why is it difficult to predict how a novel zoonotic pathogen will evolve when it spills over into humans?

### (a) Stochastic effects in small populations can overwhelm selection

Because emerging zoonotic diseases are maladapted to human populations, we certainly expect for selection to favour improved pathogen fitness. However, this does not necessarily mean that pathogens will adaptively evolve [10,13]. A key tenant of evolutionary theory is that selection must act through a background of stochasticity and drift to result in adaptive evolution [53]. Small population sizes mean that both stochasticity and drift are relatively strong, and therefore, the inevitably small population of infected individuals at the start of an epidemic means that stochasticity and drift are likely to overwhelm selection and determine the spread of mutants [53]. Additionally, the existence of founder effects during epidemic range expansions results in spatial stochasticity analogous to genetic drift [54]. Thus, founder effects and variation in transmission due to host behaviour and stochasticity likely determine the fate of mutants at the start of epidemics [10].

Additionally, adaptive evolution in acute, respiratory pathogens may be constrained by the small bottleneck sizes of transmission events [55]. Short infectious periods and small bottlenecks mean that it is less likely for a pathogen to have enough time within a host to generate adaptive mutations and select on those variants strongly enough for them to reach the high frequencies needed to transmit through tight bottlenecks [55]. This can impede adaptive evolution at the population level [56]. All of these stochastic factors can overwhelm selection, especially at the start of an epidemic. However, as the population size of infected individuals increases or if there are mutations of large enough effect size, the balance between selection and stochasticity may shift towards selection and result in adaptive evolution.

### (b) Maladapted emerging zoonotic pathogens can evolve in unexpected ways

There are many ways that emerging zoonotic pathogens can adapt to human hosts and the foremost is to improve their $R_0$ [57]. Classic trade-off theory assumes that $R_0$ should be maximized at intermediate virulence and transmission rates if these traits have tight, positive, and saturating correlations. However, these tight correlations assume that the pathogen is already relatively adapted to its host such that all potential adaptive mutations (for higher transmission rate or lower virulence) have costs (of higher virulence or lower transmission rate, respectively). This is unlikely to be the case for emerging zoonotic pathogens [40].

The concept of Pareto fronts describes scenarios where phenotypes can be in the region of sub-optimal phenotype space below the trade-off front (figure 3) [58]. The trade-off front (or Pareto front) separates these accessible, maladapted phenotype combinations from impossible, ideal phenotypes [58,59]. At the Pareto front, the two phenotypes trade-off

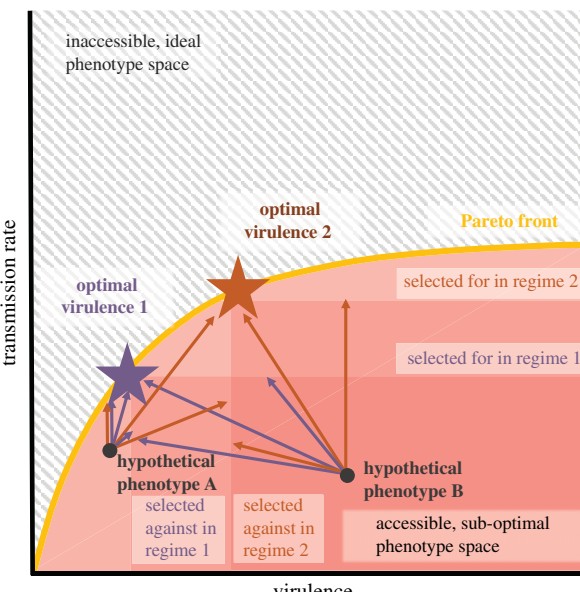

**Figure 3.** Conceptual diagram of the Pareto front between virulence and transmission rate. A Pareto front between virulence and transmission rate defines a region of accessible phenotype space. Theory determines where the 'optimal strategy' sits on the Pareto front to determine which regions of this phenotype space are selectively advantaged or disadvantaged. Phenotype combinations far from the Pareto front may technically be possible but would be highly selectively disadvantaged and likely to go extinct. Possible phenotypes can move towards their optimal strategy along any pathway within the accessible phenotype space. However, we cannot know where a hypothetical phenotype sits below its individual Pareto front. Selection for improved transmission rate can, therefore, involve decreases, no changes, or increases in virulence depending on the pathogen's starting point and mutational availability. (Online version in colour.)

with each other. Below the Pareto front, however, improvements in one trait may not affect the other trait as simple adaptations can be made before costs are incurred. Therefore, Pareto fronts determine which phenotype combinations are possible, and selection acts upon these possible phenotypes to move them towards more selectively advantageous regions.

Because they lack any evolutionary history with humans, emerging zoonotic diseases are unlikely to have fixed all available 'no-cost' adaptations and thus likely have phenotypes below Pareto fronts (figure 4). Applied to virulence evolution, this means that zoonotic diseases emerging with lower than optimal transmission rates or higher than optimal virulence may initially select for no-cost improvements even if their 'optimal' phenotype is regulated by trade-offs (figure 3) [18]. This means that, in addition to not being able to precisely predict the direction of selection because we do not know where a pathogen's starting point phenotypes sit relative to their optimal phenotypes, we cannot predict how any individual mutation improving transmission rate will affect virulence in a maladapted pathogen that starts below the Pareto front.

## 6. How does a pathogen's optimal transmission rate and virulence depend on epidemiological characteristics and change over time?

While we cannot predict exactly where the virulence and transmission rate of an emerging zoonotic disease sit relative to its

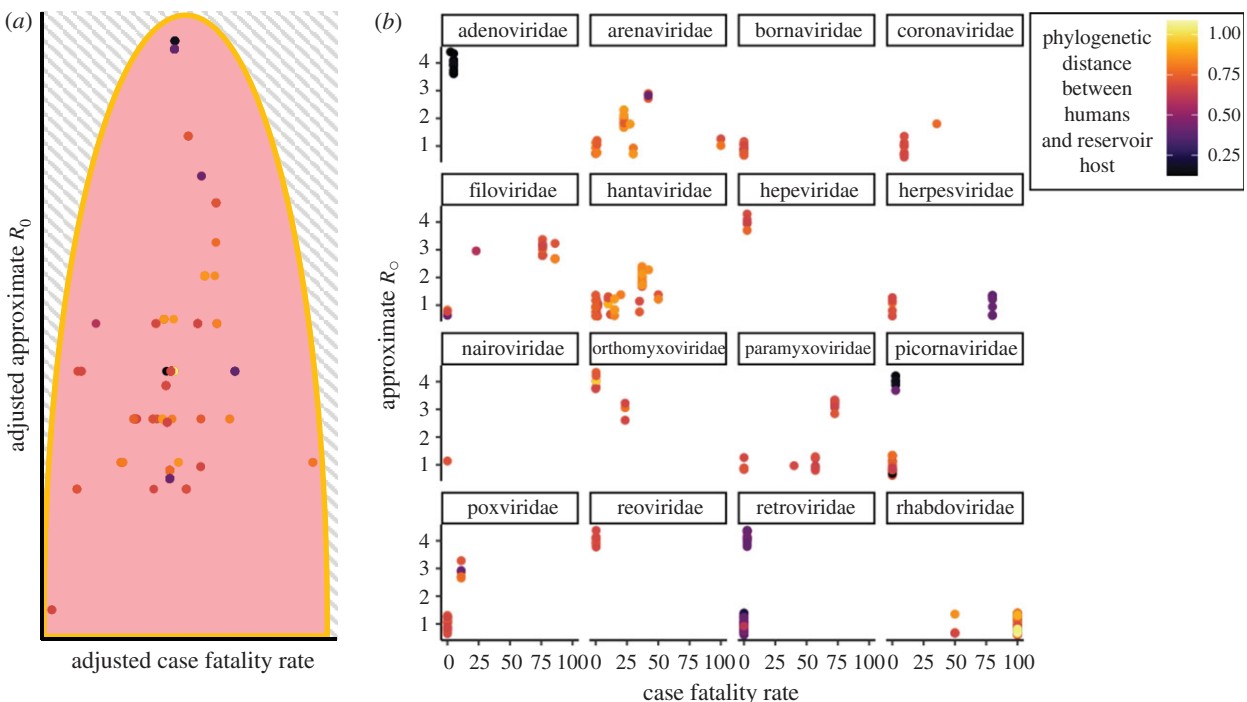

**Figure 4.** Recently emerged viral zoonoses loosely follow a Pareto front of virulence and $R_0$ where $R_0$ seems to be maximized at intermediate case fatality rates (CFRs) within viral families. Data are from a dataset published in 2019 of recently emerged viral zoonoses from mammalian hosts [40]. Approximate $R_0$ is classified from 1 (no human-to-human transmission) to 4 (endemic transmission). In figure 4a, dots represent plotted residuals from linear models of CFR and approximate $R_0$ including virus family as a factor. By regressing out virus family, we somewhat control for the variation in trade-off shape for each virus and can make general observations across the dataset. Each dot, therefore, represents the virulence and $R_0$ of an individual epidemic of viral zoonosis scaled by virus family. In figure 4b, CFR and approximate $R_0$ are directly plotted and separated by virus family so that the non-aggregated trends could be seen within virus families. In both panels, dots are coloured by the phylogenetic distance between humans and the reservoir host. Plots were made with 'ggplot2'. See electronic supplementary material for code. (Online version in colour.)

Pareto front and thus also cannot predict whether fitness-improving mutations necessarily have costs, evolutionary epidemiology theory can tell us how different epidemiological characteristics shift which regions of the possible phenotype space are selectively advantageous. Additionally, while novel zoonotic pathogens sitting far below their Pareto front may initially have costless fitness-improving mutations, their evolution will be increasingly constrained by trade-offs as their fitness improves and they approach their Pareto front.

Thus, evolutionary epidemiology theory based upon the virulence and transmission trade-off can tell us what scenarios might select for different pathogen virulence and transmission rates. However, evolutionary epidemiology theory on the virulence and transmission trade-off is perhaps more nuanced than commonly appreciated. We have discussed how variations in trade-off shape can lead to different optimal phenotypes for different pathogens [12,17,21], but the optimal values of these rates can also depend on host and parasite epidemiological characteristics and change over time in an epidemic [4,12]. While saturating virulence and transmission rate trade-offs generally predict that intermediate virulence and transmission rate is optimal, certain epidemiological characteristics can bias a system towards selecting for higher transmission rate or less virulence depending on the relative selective importance of either trait. Below, we will discuss several bodies of theory that explore how different epidemiolocal characteristics affect optimal virulence and transmission rate, specifically focusing on those where the effect of the epidemiological characteristic being explored varies depending on the time scale of the epidemic. There are also several additional sections in the supplement on these effects in systems with

multiple infection, environmental transmission (curse of the pharaoh), and antigenic escape (electronic supplementary material, S6a–c, and table S1).

## (a) Selection favours high transmission rates when susceptible density is high at the start of an epidemic

Classic models for virulence evolution examine long-term evolutionary outcomes at equilibrium [60]. Selection on virulence and transmission rates during the start of an epidemic can be explored by using models that do not assume equilibrium [18,19,41,61,62]. These models allow for the existence of multiple simultaneous mutants so that the competitive fitness of each can be assessed over shifting epidemiological conditions in time. They show that strains with higher transmission rates and virulence can be selected during epidemic growth stages, despite $R_0$ optimized (intermediate virulence) strains dominating at endemic equilibrium [19,41]. This is because strains with higher transmission rates spread fastest at the start of the epidemic when the density of susceptible hosts is high [19,41].

Intuitively, these results can be explained as: an infected host during the early stages of an epidemic encounters mostly susceptible hosts, so strains with higher transmission rates will have faster growth rates since they have shorter serial intervals (or infection generation times) than strains with higher $R_0$ (but lower transmission rates) that produce more secondary infections over a longer infectious period but more slowly. For a simplified numeric example, a strain

that has an infectious period of two days and infects 50% of its two contacts per day in an entirely susceptible population will only produce two new infections, but will double every two days. Comparatively, a strain that has an infectious period of five days and infects 40% of its two contacts per day in an entirely susceptible population will produce four new infections, but only double every 2.5 days. Thus, the higher transmission rate strain can spread faster while susceptible host densities are high during epidemic growth stages, but the $R_0$ optimized strain can outcompete it when susceptible density is low at endemic equilibrium because it produces a larger number of infections over its longer infectious period. Therefore, improvements in transmission rate are the most important at the start of an epidemic and can be selected for even if they have shorter infectious periods due to increased virulence. This also demonstrates that the high density of susceptible hosts early in epidemics crucially influences selection [12,18,19,41].

## (b) Structured host populations select for prudent strategies at equilibrium, but transiently select for virulent strategies at the epidemic front

Classic virulence evolution trade-off theory assumes that transmission happens randomly in a homogeneously mixing population [12]. However, natural populations almost always have heterogeneous mixing patterns due to spatial structure and social networks [63,64]. In these structured populations, transmission occurs more often between neighbouring individuals and those in social groups. This can lead to 'self-shading' where highly infectious strains rapidly deplete their local susceptible populations and compete for available hosts with related strains [63,65]. Thus, structured host populations select for lower pathogen infectivity and virulence at endemic equilibrium. However, the high availability of susceptible hosts at the start of an epidemic is likely to reduce the impact of self-shading and, moreover, pathogens need to have higher transmission rates to seed an epidemic in a spatially structured population than in a well-mixed one [66]. Before equilibrium, the invasion front of a spatially structured epidemic also has a high local supply of susceptible hosts, which leads to a dynamic where virulent, high transmission rate strains are selected at the invasion front and then are succeeded by more prudent strategies as the local dynamics approach equilibrium [67,68]. Overall, then, it is possible that structure in host populations temporarily selects for higher virulence while the epidemic is spreading through mostly susceptible populations. However, if there are also trade-offs where high virulence impedes host movement, then the spatial front of the epidemic might instead have lower virulence [69]. As such, it is unclear how population structure and movement overall will select emerging pathogens during different parts of the epidemic.

## (c) How might public health measures shape selection on virulence and transmission rate?

The question of whether public health measures can purposely or inadvertently drive pathogen evolution naturally arises when discussing virulence evolution. Public health measures intentionally driving the evolution of virulence may be unrealistic in emerging zoonotic diseases because,

as we have discussed, virulence evolution is very difficult to fully predict [9]. However, we can gain insight into how public health measures can inadvertently select on virulence. Non-pharmaceutical public health interventions for epidemics primarily aim to decrease transmission and, therefore, either stop the epidemic or slow it until vaccines and treatments can be developed. This decreases the total number of infected individuals, which will have the greatest impact on the total mortality burden of any epidemic [7]. This also limits the evolutionary potential of the pathogen by limiting the number of cases and, therefore, the strength of selection and opportunities for mutation [7]. However, some of these interventions may also contribute to the selection acting on the pathogen [7,9]. First, decreased travel and extra-household contacts should alter the spatial and social structure of the population to make a more structured transmission network, which might prevent low transmission rate pathogens from spreading initially [63,66]. Second, quarantine of symptomatic individuals may select for decreased or altered symptoms, which could select for lower virulence if symptoms are linked to virulence [70]. Third, increased environmental sanitation decreases environmental transmission, thus potentially selecting for altered pathogen virulence under the 'curse of the pharaoh' hypothesis [71] (see electronic supplementary material, S6b). Finally, vaccines can sometimes create selection pressures on pathogens with potential evolutionary impacts to consider [72] (see electronic supplementary material, S6c).

While the most human mortality will be prevented by simply preventing transmission, considering the effects of control measures on pathogen evolution can, in principle, lead to better epidemic management [7]. Understanding host population characteristics creating strong selection for high transmission rate strategies could help distribute public health effort if there are limited resources [7]. However, a key point is that weak epidemic control measures that allow for extended transmission in humans increase the evolutionary potential of zoonotic pathogens because they allow for stronger selection and more mutations [7]. Thus, the best evolutionary management practice for an epidemic of a zoonotic infectious disease would be to suppress transmission using strong, rapid public health interventions.

## 7. Conclusion

In the face of the extraordinarily stressful circumstances of a global pandemic, we all understandably want simple answers for what will happen next and how the pathogen will evolve. Unfortunately, the simplest answer is that we cannot predict the evolution of any specific novel zoonotic pathogen. Its virulence and transmission rate may trade-off; it may be selected to increase its transmission rate; and the dynamics of selection may change with time.

The slightly more complicated answer is that, while we cannot predict how any specific pathogen will evolve, we do know how selection is expected to generally act on emerging zoonotic diseases and how different assumptions affect these predictions. We know that novel zoonotic pathogens emerge into the human population maladapted to human hosts [40,50]. Generally, we expect that virulence and transmission rate trade-off, leading to selection towards intermediate values of both [17]. However, we also know that a maladapted

zoonotic pathogen's virulence and transmission phenotypes may start below the Pareto front, so selection for higher transmission rates can have decoupled effects on virulence [18]. Our theory also says that, with trade-offs, the optimal balance between virulence and transmission rate shifts depending on the time scale of the epidemic and different epidemiological and population characteristics [17,18].

All of these uncertainties make virulence evolution an academically interesting topic with a rich body of theory surrounding it, but no universal predictions [9]. Unfortunately, any sort of evolutionary prediction depends on a good understanding of how the phenotypes that the pathogen emerges with compare to their 'optimal' phenotypes in human populations; what fitness-improving mutations the pathogen has available to it and what their associated trade-offs are; and how host population structure and epidemiological characteristics will shape the selection pressures on the pathogen. These data are exceptionally difficult to quickly gather. However, despite our inability to conclusively predict how a pathogen will evolve, we do know that we can prevent it from doing so by implementing strong, rapid public health measures that suppress transmission early on since this will decrease the evolutionary potential of such pathogens while also decreasing the total mortality burden by limiting the number of people infected.

Data accessibility. No novel data are used in this manuscript; data used are publicly available as online electronic supplementary material from [40]. The annotated R script used for analysis is provided in the electronic supplementary material.

Authors' contributions. All authors researched and edited the paper. E.V. and M.B. conceptualized and wrote the paper.

Competing interests. We declare we have no competing interests.

Funding. E.V., S.G., and N.A.S. acknowledge funding from NSF GRFP DGE 1752814 grants. E.V. and M.N. acknowledge funding from the UC Berkeley SURF-SMART program. E.V. acknowledges funding from the Philomathia Foundation Graduate Student Fellowship in the Environmental Sciences. E.V. and M.B. acknowledge funding from NSF DEB 2011109. C.R. and M.B. acknowledge funding from NIH/R01-GM122061-03. M.B. acknowledges funding from NERC NE/V012347/1.

Acknowledgements. We thank members of the Boots lab for helpful discussions and several reviewers for their very useful comments. We also thank the many virologists and evolutionary biologists on twitter whose threads on viral evolution helped shaped this paper.

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
