## [Peer Review File · Proceedings of the Royal Society B: Biological Sciences]

Review History

RSPB-2021-0900.R0 (Original submission)

Review form: Reviewer 1

Recommendation

Major revision is needed (please make suggestions in comments)

Scientific importance: Is the manuscript an original and important contribution to its field?

Good

General interest: Is the paper of sufficient general interest?

Excellent

Quality of the paper: Is the overall quality of the paper suitable?

Acceptable

Is the length of the paper justified?

Yes

Should the paper be seen by a specialist statistical reviewer?

No

Do you have any concerns about statistical analyses in this paper? If so, please specify them explicitly in your report.

No

It is a condition of publication that authors make their supporting data, code and materials available - either as supplementary material or hosted in an external repository. Please rate, if applicable, the supporting data on the following criteria.

Is it accessible?

N/A

Is it clear?

N/A

Is it adequate?

N/A

Do you have any ethical concerns with this paper?

No

Comments to the Author

Review of Visher et al. "The Three Ts of Virulence Evolution During Zoonotic Emergence" for PRSB

General comments:

I found this review well-written and easy to read, the 3-T framework and the introduction compelling, the topic of general interest, and Figures 1 and 3 very interesting/useful. My main concerns, detailed below, are that I thought Table 1 was confusing (in terms of the topic areas and specific studies selected and why) and that the review could benefit from strengthening the connections among the framework, Table 1, and the second half of the review. I also at times had trouble identifying if the traits being discussed were at the scale of the pathogen population or the host population.

Detailed comments:

I felt there was a disconnect between the introduction/framework, the details in Table 1, and the sections presented later in the review. Strengthening the connections among these would make this review more effective. I recognize the challenges of condensing such a broad literature into a short review. But clarifying in the introductory sections why these specific topics are explored would help focus our attention. Are these a laundry list of questions people have been asking about the current pandemic? Or specific topics that aren't typically discussed, but should be, because they are known to affect the observed dynamics in other systems? Or both?

From the introduction, I expected the sections that followed to mirror the 4 questions presented at lines 74-78. Then I was a bit confused at why some of the sections that followed were there since they weren't motivated earlier (e.g., multiple infection, structured populations, environmental transmission, antigenic escape). One way to merge these would be to have main sections that follow the 4 questions noted in 74-78, with subsections within that address critical topics that alter expected outcomes for that main question. I also found section headings framed as questions more compelling than those that were not. Could all the subsections after the framework be framed as specific questions we might/should be asking about pathogen evolution in the SARS-CoV-2 pandemic?

I then expected the topics presented in Table 1 to reflect the sections addressed in the review. This was partially true, but there were additional sections than just those topics presented in Table 1.

Table 1 could use some explanation about what/why these examples were selected. These examples and topics are a portion of empirical tests of evolutionary theory – why/how were these chosen? It's also not clear how some of these examples fit the trade-off given the strict definition of virulence given in Box 1 (=parasite-induced increase in host mortality rate). E.g., how does the Acevedo review support the trade-off when both virulence and transmission have an increasing relationship with replication rate, and not to each other?

More on table 1 topics: How about empirical studies of virulence or transmission at spillover/introduction vs once established in a population? That seems a topic more relevant to the review, and there are examples from emerging diseases in wildlife (some examples that come to mind include devil facial tumor disease, white-nose syndrome, myxoma virus, west Nile).

Also, text on lines 143 refers to table 1 as evidence of “an increasing number of empirical studies finding support for the core idea” yet not all of these studies support the trade-off hypothesis (as one example, Walther and Ewald), and the idea that support is “increasing” is not evident in the table.

209-231: this section is great, and I learned something new!

237-238. The point that classic models of evolutionary trade-offs between virulence and evolution are evaluated at equilibrium is an important one, and could occur earlier in the paper, either when introducing the 3-Ts or in the trade-off hypothesis section.

241-245: consider replacing “strategies” with “mutants” or “strains”

L246-253: this paragraph needs clarification. I don't see an intuitive link between mutants with a diversity of transmission rates as discussed in the previous paragraph and the relationships between transmission, generation time, and infectious period early in an epidemic as discussed here. All else being the same (length of generations and infectious periods being equal), a mutant with a higher transmission rate (a higher R_0) will lead to more secondary infections than one with a lower transmission rate (a lower R_0), just based on the larger number of infectious contacts and their subsequent contacts holding a higher proportion of the total infected population as the epidemic grows. I don't see how transmission rate here in any way relates to variation in the length of the infectious period or to generation time. This paragraph needs more explanation to make this connection.

The arguments presented earlier in this section (“selection on virulence and transmission rate during epidemics”), and in Figure 3a's legend, suggest that early in the epidemic “selection for improved transmission rate can therefore involve decreases, no changes, or increases in virulence depending on the pathogen's starting point and mutational availability.” Thus, the suggestion on lines 247-250 that in the early stages of the epidemic “strains with higher transmission rates will have faster population growth rates since they have shorter generation times than strains with lower transmission rates...” is confusing to me. Part of my confusion here may be from a lack of clarity in what is meant by generation time (the infection moving from one host to another host? Or a pathogen generation within an infected host?), and what is meant by population growth rate (the replication rate of the pathogen in a host? Or the rate of increase in the number of infected hosts?). But either way I consider the meanings here, I still don't see an intuitive link that increased transmission rate at this stage necessarily links to generation time or duration of infectiousness.

255-264: this section is vague. More explanation/examples would help. How does virulence change with multiple infection? What sort of mechanisms for competition or virulence will make what sort of differences in virulence or transmission rate?

261-263: Does prevalence need to be high, and does the epidemic need to be more developed, for multiple infections to occur? Can't diversity be generated within a host's pathogen

population during an infection, and then a diverse infection spread?

The section at 255-264 could possibly be combined with the next section on structured host populations? Both deal with population structure (of pathogen or host), and how structure alters evolutionary outcomes. Also, the effect of pathogen diversity is discussed briefly in the host population structure section. Some more information about how pathogen diversity would interact with host structure at different stages of the epidemic would be useful (if known).

308-311: the authors briefly touch on selection for antigenic or vaccine escape, and note they will not go into more detail on that topic. If this could be expanded at least a bit, that would be very useful, given that this topic is of great interest to the SARS-CoV-2 pandemic response.

333-337: these sentences note how public health measures may alter contacts, spatial structure, symptoms and selection. But they don't go the next step to discuss how those changes could in turn affect virulence or transmission, which are the goals of the review. I understand not wanting to make specific predictions, but for the first point (lines 331-333) you did note how the response might change virulence, and then I was expecting a similar structure to the rest of these, wherein you discussed how each intervention could affect transmission or virulence (and in which direction).

370: Edit to "these data are..."

Review form: Reviewer 2

Recommendation

Major revision is needed (please make suggestions in comments)

Scientific importance: Is the manuscript an original and important contribution to its field?

Acceptable

General interest: Is the paper of sufficient general interest?

Good

Quality of the paper: Is the overall quality of the paper suitable?

Excellent

Is the length of the paper justified?

Yes

Should the paper be seen by a specialist statistical reviewer?

No

Do you have any concerns about statistical analyses in this paper? If so, please specify them explicitly in your report.

No

It is a condition of publication that authors make their supporting data, code and materials available - either as supplementary material or hosted in an external repository. Please rate, if applicable, the supporting data on the following criteria.

Is it accessible?

Yes

Is it clear?

Yes

Is it adequate?

Yes

Do you have any ethical concerns with this paper?

No

Comments to the Author

In this paper, the Visher et al. review existing theoretical literature to further our understanding of the evolution of emerging infectious diseases. There were many things that I liked about this paper. In particular, I found it incredibly easy and enjoyable to read - the authors have a distinctive "voice" that is clear and compelling. I also found Fig. 1 to be very useful. But my major compliment is with the inclusion of the discussion of Pareto fronts - when I started reviewing this paper, I wondered whether I would learn anything new, given existing reviews of virulence evolution and papers studying virulence evolution under non-equilibrium conditions: the Pareto front discussion was novel to me, and thus a very welcome and productive addition to the paper. I do have several comments and suggestions that I think would improve the paper.

As noted above, there have been many primary literature articles and reviews that cover seemingly similar ground to your review (e.g., Bull & Ebert 2008; Bolker et al. 2010 *J. R. Soc. Interface* 7, 811-822; Cressler et al. 2016; Bonneaud & Longdon 2020). It would be helpful to the reader if there was a clear statement that explains the unique contribution of this paper is by setting it in the context of these seemingly similar papers. For example, you state (lines 71-74), "our aim is to provide a framework so that readers understand the general principles of pathogen virulence and transmission evolution and can also see how variations in the assumptions of these models based upon nuances of biology and population structure can lead to deviations in their predictions." How is that aim different from the aims of previous papers?

Another set of major comments concern Fig. 3. I really like the idea of thinking about virulence and transmission evolution via Pareto fronts, as this was not an idea I had encountered before. I actually think you could probably expand on the discussion in this section a bit, as it seemed like there was scope to discuss how the idea of maladaptation might interact with other factors (especially stochasticity and small population sizes) to govern what portion of accessible, sub-optimal phenotype space is actually achievable. That is, it seems like it should be possible to also bound the accessible phenotype space of Fig. 3a from below with biologically infeasible transmission/virulence combinations that are unlikely to survive the highly stochastic early events of spillover (e.g., high virulence, low transmission rate pathogens such as HPAI).

I also appreciated the attempt to use existing data on human zoonoses in Fig. 3b. However, I feel like there are a lot of unstated assumptions going into Fig. 3b that need to be made explicit, and in so doing may reduce the viability of that figure. In particular, I am uncomfortable with the idea of drawing a single Pareto front for all of these zoonoses. Drawing it this way suggests some kind of universal Pareto front (and thus a universal transmission-virulence relationship) that applies to all emerging infections. But, as you emphasize throughout your paper, the details of this trade-off are likely to be highly system-specific so any universal Pareto front is unlikely to exist.

However, I wonder if there is a way to examine this idea a bit more quantitatively. As you (ref. 27) and other authors (refs. 53 and 56; also a citation for Farrell and Davies 2018 *PNAS* 116, 7911 would be appropriate here) have noted, the phylogenetic distance between the reservoir host and the spillover host is an important determinant of transmission, virulence, R_0 , and general "maladaptedness" of pathogens. Thus, I would expect that there should be a relationship between the phylogenetic distance between humans and the reservoir host and the distance to your hypothetical Pareto front. If that is the case, then the far right point on your plot should represent a pathogen that has spilled over from a very close relative, whereas the points near the

middle should represent spillovers from more phylogenetically distant hosts. If there is such a relationship, then that would increase my confidence that this universal Pareto front might actually exist, rather than just being drawn on as a visual aid that might not pass biological muster.

In the section, "Selection on virulence and transmission rate during epidemics" (and really all of the sections that follow this section), it would be helpful to explain how the interpretation of these papers is affected by the notion of Pareto fronts and no-cost mutations. The difficulty I am having with these sections is that they follow two sections where you basically suggest that trade-offs might not matter very much during spillover (because of stochasticity and no-cost mutations in initially maladapted pathogens), but all of these sections are discussing results that have come out of theoretical papers that have studied virulence evolution in a trade-off context. I found myself having a hard time thinking about when and how evolution might be guided by the predictions of existing theory in these section, given what I had just read about stochasticity and no-cost mutations. Maybe this could be handled by a reordering of the sections? I.e., would it make sense to start the discussion with lines 232-317, and then step back and say, "All of the preceding was focused on evolution in context where stochasticity is unimportant and evolution is constrained by trade-offs. These situations may be quite rare, in general for emerging pathogens." And so on. I'm not sure if that's better, but I thought I'd throw it out there for your consideration.

Minor comments:

Lines 61-63: Can you explain what you mean a bit more here? There have been a number of good reviews of the evolution of virulence that try to capture the nuances of evolution in emerging zoonotic disease in a single place. (This is related to my first comment about what the unique contribution of this paper is to be.)

Line 96: Given that you are discussing mathematical models, I think it's a bit confusing to use "in theory" in this sentence, since you are using it to mean "potentially" rather than "in a mathematical model."

Box 1: "Together the transmission rate and duration of infectiousness (the inverse of virulence) determine the pathogens R_0 ..." It is inaccurate to say that duration of infectiousness is the inverse of virulence, since duration of infectiousness is also determined by other things (in particular, recovery). Overall, I did not find Box 1 to be particularly useful and think it could be cut.

Lines 189-190: I know what you mean by "this does not necessarily mean that there will be adaptive evolution," but it is a bit jarring with the preceding sentence that says that "we expect for there to be selection for improved pathogen fitness," since "adaptive evolution" IS "selection for improved fitness." Maybe restate as "this does not necessarily mean that adaptive evolution will be the primary determinant of the evolutionary trajectory of pathogen traits," or something like that.

Decision letter (RSPB-2021-0900.R0)

25-May-2021

Dear Dr Visher:

Your manuscript has now been peer reviewed and the reviewers' comments (not including confidential comments to the Editor) are included at the end of this email for your reference. As you will see, the reviewers think your paper has the potential to be an excellent contribution, but

not without some important revisions. I agree. What's needed is explained in some detail below, and the fact that the reviewers have done such a thorough job is testimony to their enthusiasm for the topic and your treatment of it. So, I would like to invite you to revise your manuscript to address their concerns.

We do not allow multiple rounds of revision so we urge you to make every effort to fully address all of the comments at this stage. If deemed necessary, your manuscript will be sent back to one or more of the original reviewers for assessment. If the original reviewers are not available we may invite new reviewers. Please note that we cannot guarantee eventual acceptance of your manuscript at this stage.

Research ethics:

Use of animals and field studies:

It is a condition of publication that you make available the data and research materials supporting the results in the article (<https://royalsociety.org/journals/authors/author-guidelines/#data>). Datasets should be deposited in an appropriate publicly available repository and details of the associated accession number, link or DOI to the datasets must be included in the Data Accessibility section of the article (<https://royalsociety.org/journals/ethics-policies/data-sharing-mining/>). Reference(s) to datasets should also be included in the reference list of the article with DOIs (where available).

Please submit a copy of your revised paper within three weeks. If we do not hear from you within this time your manuscript will be rejected. If you are unable to meet this deadline please let us know as soon as possible, as we may be able to grant a short extension.

Best wishes,
Innes Cuthill

Prof. Innes Cuthill
Reviews Editor, Proceedings B
mailto: proceedingsb@royalsociety.org

Reviewer(s)' Comments to Author:

Referee: 1

Comments to the Author(s)

Review of Visher et al. "The Three Ts of Virulence Evolution During Zoonotic Emergence" for PRSB

General comments:

I found this review well-written and easy to read, the 3-T framework and the introduction compelling, the topic of general interest, and Figures 1 and 3 very interesting/useful. My main concerns, detailed below, are that I thought Table 1 was confusing (in terms of the topic areas and specific studies selected and why) and that the review could benefit from strengthening the connections among the framework, Table 1, and the second half of the review. I also at times had trouble identifying if the traits being discussed were at the scale of the pathogen population or the host population.

Detailed comments:

I felt there was a disconnect between the introduction/framework, the details in Table 1, and the sections presented later in the review. Strengthening the connections among these would make this review more effective. I recognize the challenges of condensing such a broad literature into a short review. But clarifying in the introductory sections why these specific topics are explored

would help focus our attention. Are these a laundry list of questions people have been asking about the current pandemic? Or specific topics that aren't typically discussed, but should be, because they are known to affect the observed dynamics in other systems? Or both?

From the introduction, I expected the sections that followed to mirror the 4 questions presented at lines 74-78. Then I was a bit confused at why some of the sections that followed were there since they weren't motivated earlier (e.g., multiple infection, structured populations, environmental transmission, antigenic escape). One way to merge these would be to have main sections that follow the 4 questions noted in 74-78, with subsections within that address critical topics that alter expected outcomes for that main question. I also found section headings framed as questions more compelling than those that were not. Could all the subsections after the framework be framed as specific questions we might/should be asking about pathogen evolution in the SARS-CoV-2 pandemic?

I then expected the topics presented in Table 1 to reflect the sections addressed in the review. This was partially true, but there were additional sections than just those topics presented in Table 1.

Table 1 could use some explanation about what/why these examples were selected. These examples and topics are a portion of empirical tests of evolutionary theory – why/how were these chosen? It's also not clear how some of these examples fit the trade-off given the strict definition of virulence given in Box 1 (=parasite-induced increase in host mortality rate). E.g., how does the Acevedo review support the trade-off when both virulence and transmission have an increasing relationship with replication rate, and not to each other?

More on table 1 topics: How about empirical studies of virulence or transmission at spillover/introduction vs once established in a population? That seems a topic more relevant to the review, and there are examples from emerging diseases in wildlife (some examples that come to mind include devil facial tumor disease, white-nose syndrome, myxoma virus, west Nile).

Also, text on lines 143 refers to table 1 as evidence of "an increasing number of empirical studies finding support for the core idea" yet not all of these studies support the trade-off hypothesis (as one example, Walther and Ewald), and the idea that support is "increasing" is not evident in the table.

209-231: this section is great, and I learned something new!

237-238. The point that classic models of evolutionary trade-offs between virulence and evolution are evaluated at equilibrium is an important one, and could occur earlier in the paper, either when introducing the 3-Ts or in the trade-off hypothesis section.

241-245: consider replacing "strategies" with "mutants" or "strains"

L246-253: this paragraph needs clarification. I don't see an intuitive link between mutants with a diversity of transmission rates as discussed in the previous paragraph and the relationships between transmission, generation time, and infectious period early in an epidemic as discussed here. All else being the same (length of generations and infectious periods being equal), a mutant with a higher transmission rate (a higher R_0) will lead to more secondary infections than one with a lower transmission rate (a lower R_0), just based on the larger number of infectious contacts and their subsequent contacts holding a higher proportion of the total infected population as the epidemic grows. I don't see how transmission rate here in any way relates to variation in the length of the infectious period or to generation time. This paragraph needs more explanation to make this connection.

The arguments presented earlier in this section ("selection on virulence and transmission rate during epidemics"), and in Figure 3a's legend, suggest that early in the epidemic "selection for improved transmission rate can therefore involve decreases, no changes, or increases in virulence depending on the pathogen's starting point and mutational availability." Thus, the suggestion on

lines 247-250 that in the early stages of the epidemic “strains with higher transmission rates will have faster population growth rates since they have shorter generation times than strains with lower transmission rates...” is confusing to me. Part of my confusion here may be from a lack of clarity in what is meant by generation time (the infection moving from one host to another host? Or a pathogen generation within an infected host?), and what is meant by population growth rate (the replication rate of the pathogen in a host? Or the rate of increase in the number of infected hosts?). But either way I consider the meanings here, I still don't see an intuitive link that increased transmission rate at this stage necessarily links to generation time or duration of infectiousness.

255-264: this section is vague. More explanation/examples would help. How does virulence change with multiple infection? What sort of mechanisms for competition or virulence will make what sort of differences in virulence or transmission rate?

261-263: Does prevalence need to be high, and does the epidemic need to be more developed, for multiple infections to occur? Can't diversity be generated within a host's pathogen population during an infection, and then a diverse infection spread?

The section at 255-264 could possibly be combined with the next section on structured host populations? Both deal with population structure (of pathogen or host), and how structure alters evolutionary outcomes. Also, the effect of pathogen diversity is discussed briefly in the host population structure section. Some more information about how pathogen diversity would interact with host structure at different stages of the epidemic would be useful (if known).

308-311: the authors briefly touch on selection for antigenic or vaccine escape, and note they will not go into more detail on that topic. If this could be expanded at least a bit, that would be very useful, given that this topic is of great interest to the SARS-CoV-2 pandemic response.

333-337: these sentences note how public health measures may alter contacts, spatial structure, symptoms and selection. But they don't go the next step to discuss how those changes could in turn affect virulence or transmission, which are the goals of the review. I understand not wanting to make specific predictions, but for the first point (lines 331-333) you did note how the response might change virulence, and then I was expecting a similar structure to the rest of these, wherein you discussed how each intervention could affect transmission or virulence (and in which direction).

370: Edit to “these data are...”

Referee: 2

Comments to the Author(s)

In this paper, the Visher et al. review existing theoretical literature to further our understanding of the evolution of emerging infectious diseases. There were many things that I liked about this paper. In particular, I found it incredibly easy and enjoyable to read - the authors have a distinctive "voice" that is clear and compelling. I also found Fig. 1 to be very useful. But my major compliment is with the inclusion of the discussion of Pareto fronts - when I started reviewing this paper, I wondered whether I would learn anything new, given existing reviews of virulence evolution and papers studying virulence evolution under non-equilibrium conditions: the Pareto front discussion was novel to me, and thus a very welcome and productive addition to the paper. I do have several comments and suggestions that I think would improve the paper.

As noted above, there have been many primary literature articles and reviews that cover seemingly similar ground to your review (e.g., Bull & Ebert 2008; Bolker et al. 2010 *J. R. Soc. Interface* 7, 811-822; Cressler et al. 2016; Bonneaud & Longdon 2020). It would be helpful to the reader if there was a clear statement that explains the unique contribution of this paper is by setting it in the context of these seemingly similar papers. For example, you state (lines 71-74),

“our aim is to provide a framework so that readers understand the general principles of pathogen virulence and transmission evolution and can also see how variations in the assumptions of these models based upon nuances of biology and population structure can lead to deviations in their predictions.” How is that aim different from the aims of previous papers?

Another set of major comments concern Fig. 3. I really like the idea of thinking about virulence and transmission evolution via Pareto fronts, as this was not an idea I had encountered before. I actually think you could probably expand on the discussion in this section a bit, as it seemed like there was scope to discuss how the idea of maladaptation might interact with other factors (especially stochasticity and small population sizes) to govern what portion of accessible, sub-optimal phenotype space is actually achievable. That is, it seems like it should be possible to also bound the accessible phenotype space of Fig. 3a from below with biologically infeasible transmission/virulence combinations that are unlikely to survive the highly stochastic early events of spillover (e.g., high virulence, low transmission rate pathogens such as HPAI).

I also appreciated the attempt to use existing data on human zoonoses in Fig. 3b. However, I feel like there are a lot of unstated assumptions going into Fig. 3b that need to be made explicit, and in so doing may reduce the viability of that figure. In particular, I am uncomfortable with the idea of drawing a single Pareto front for all of these zoonoses. Drawing it this way suggests some kind of universal Pareto front (and thus a universal transmission-virulence relationship) that applies to all emerging infections. But, as you emphasize throughout your paper, the details of this trade-off are likely to be highly system-specific so any universal Pareto front is unlikely to exist.

However, I wonder if there is a way to examine this idea a bit more quantitatively. As you (ref. 27) and other authors (refs. 53 and 56; also a citation for Farrell and Davies 2018 PNAS 116, 7911 would be appropriate here) have noted, the phylogenetic distance between the reservoir host and the spillover host is an important determinant of transmission, virulence, R_0 , and general “maladaptedness” of pathogens. Thus, I would expect that there should be a relationship between the phylogenetic distance between humans and the reservoir host and the distance to your hypothetical Pareto front. If that is the case, then the far right point on your plot should represent a pathogen that has spilled over from a very close relative, whereas the points near the middle should represent spillovers from more phylogenetically distant hosts. If there is such a relationship, then that would increase my confidence that this universal Pareto front might actually exist, rather than just being drawn on as a visual aid that might not pass biological muster.

In the section, “Selection on virulence and transmission rate during epidemics” (and really all of the sections that follow this section), it would be helpful to explain how the interpretation of these papers is affected by the notion of Pareto fronts and no-cost mutations. The difficulty I am having with these sections is that they follow two sections where you basically suggest that trade-offs might not matter very much during spillover (because of stochasticity and no-cost mutations in initially maladapted pathogens), but all of these sections are discussing results that have come out of theoretical papers that have studied virulence evolution in a trade-off context. I found myself having a hard time thinking about when and how evolution might be guided by the predictions of existing theory in these section, given what I had just read about stochasticity and no-cost mutations. Maybe this could be handled by a reordering of the sections? I.e., would it make sense to start the discussion with lines 232-317, and then step back and say, “All of the preceding was focused on evolution in context where stochasticity is unimportant and evolution is constrained by trade-offs. These situations may be quite rare, in general for emerging pathogens.” And so on. I’m not sure if that’s better, but I thought I’d throw it out there for your consideration.

Minor comments:

Lines 61-63: Can you explain what you mean a bit more here? There have been a number of good reviews of the evolution of virulence that try to capture the nuances of evolution in emerging

zoonotic disease in a single place. (This is related to my first comment about what the unique contribution of this paper is to be.)

Line 96: Given that you are discussing mathematical models, I think it's a bit confusing to use "in theory" in this sentence, since you are using it to mean "potentially" rather than "in a mathematical model."

Box 1: "Together the transmission rate and duration of infectiousness (the inverse of virulence) determine the pathogens R_0 ..." It is inaccurate to say that duration of infectiousness is the inverse of virulence, since duration of infectiousness is also determined by other things (in particular, recovery). Overall, I did not find Box 1 to be particularly useful and think it could be cut.

Lines 189-190: I know what you mean by "this does not necessarily mean that there will be adaptive evolution," but it is a bit jarring with the preceding sentence that says that "we expect for there to be selection for improved pathogen fitness," since "adaptive evolution" IS "selection for improved fitness." Maybe restate as "this does not necessarily mean that adaptive evolution will be the primary determinant of the evolutionary trajectory of pathogen traits," or something like that.

Author's Response to Decision Letter for (RSPB-2021-0900.R0)

See Appendix A.

RSPB-2021-0900.R1 (Revision)

Review form: Reviewer 1

Recommendation

Accept with minor revision (please list in comments)

Scientific importance: Is the manuscript an original and important contribution to its field?

Excellent

General interest: Is the paper of sufficient general interest?

Excellent

Quality of the paper: Is the overall quality of the paper suitable?

Excellent

Is the length of the paper justified?

Yes

Should the paper be seen by a specialist statistical reviewer?

No

Do you have any concerns about statistical analyses in this paper? If so, please specify them explicitly in your report.

No

It is a condition of publication that authors make their supporting data, code and materials available - either as supplementary material or hosted in an external repository. Please rate, if applicable, the supporting data on the following criteria.

Is it accessible?

N/A

Is it clear?

N/A

Is it adequate?

N/A

Do you have any ethical concerns with this paper?

No

Comments to the Author

Review of revision by Visher et al. "The Three Ts of Virulence Evolution During Zoonotic Emergence" for PRSB

The authors did a great job revising the manuscript, and addressed all my comments on the previous version. The revision is a stronger paper, and I can't wait for it to be published, so I can share it with colleagues. I really liked the changes to L261-274 explaining differences in transmission rates and R_0 .

I have only a few minor suggestions:

Supplemental figure 1 is great, and I hate to have it buried in the SI. Any chance this could be squeezed into the main text? Maybe it could join Figure 3 as an extra panel? (Either way, perhaps have an arrow labeling SARS-CoV-2 on that plot, assuming it's there.)

Figure 3b legend: replace "potted residuals" with "plotted residuals"

L307-309. This is perhaps a splitting of hairs, but I'm not convinced that increased environmental sanitation would select for lower virulence "under the curse of the pharaoh". Sanitation would kill propagules in the environment whether they are long-lived/more virulent or short-lived/less virulent. I can imagine if multi-mode transmission is possible, it could shift the system to more direct transmission, which could support less virulent strains than longer-lived propagules with the option of environmental transmission. But that is more about variation in virulence among transmission modes, not under the umbrella of curse of the pharaoh.

R_0 is sometimes written as R_0 , R_o , or R_0 (subscript). Standardize.

Decision letter (RSPB-2021-0900.R1)

12-Jul-2021

Dear Dr Visher

I am pleased to inform you that your manuscript RSPB-2021-0900.R1 entitled "The Three Ts of Virulence Evolution During Zoonotic Emergence" has been accepted for publication in Proceedings B.

The referee is very happy with your revisions, but also suggested a couple of minor revisions to your manuscript. Therefore, I invite you to respond to the referee's comments and upload the final version of your manuscript. Because the schedule for publication is very tight, it is a condition of publication that you submit the revised version of your manuscript within 7 days. If you do not think you will be able to meet this date please let us know.

[http://datadryad.org/submit?journalID=RSPB&manu=\(Document not available\)](http://datadryad.org/submit?journalID=RSPB&manu=(Document+not+available)) which will take you to your unique entry in the Dryad repository. If you have already submitted your data to dryad you can make any necessary revisions to your dataset by following the above link. Please see <https://royalsociety.org/journals/ethics-policies/data-sharing-mining/> for more details.

Best wishes,
Innes Cuthill

Prof. Innes Cuthill
Reviews Editor, Proceedings B
mailto: proceedingsb@royalsociety.org

Reviewer(s)' Comments to Author:

Referee: 1

Comments to the Author(s)

Review of revision by Visher et al. "The Three Ts of Virulence Evolution During Zoonotic Emergence" for PRSB

The authors did a great job revising the manuscript, and addressed all my comments on the previous version. The revision is a stronger paper, and I can't wait for it to be published, so I can share it with colleagues. I really liked the changes to L261-274 explaining differences in transmission rates and R0.

I have only a few minor suggestions:

Supplemental figure 1 is great, and I hate to have it buried in the SI. Any chance this could be squeezed into the main text? Maybe it could join Figure 3 as an extra panel? (Either way, perhaps have an arrow labeling SARS-CoV-2 on that plot, assuming it's there.)

Figure 3b legend: replace "potted residuals" with "plotted residuals"

L307-309. This is perhaps a splitting of hairs, but I'm not convinced that increased environmental sanitation would select for lower virulence "under the curse of the pharaoh". Sanitation would kill propagules in the environment whether they are long-lived/more virulent or short-lived/less virulent. I can imagine if multi-mode transmission is possible, it could shift the system to more direct transmission, which could support less virulent strains than longer-lived propagules with

the option of environmental transmission. But that is more about variation in virulence among transmission modes, not under the umbrella of curse of the pharaoh.

R0 is sometimes written as R_0 , R_o , or R_0 (subscript). Standardize.

Author's Response to Decision Letter for (RSPB-2021-0900.R1)

See Appendix B.

Decision letter (RSPB-2021-0900.R2)

16-Jul-2021

Dear Ms Visher

I am pleased to inform you that your manuscript entitled "The Three Ts of Virulence Evolution During Zoonotic Emergence" has been accepted for publication in Proceedings B.

If you are likely to be away from e-mail contact during this period, let us know. Due to rapid publication and an extremely tight schedule, if comments are not received, we may publish the paper as it stands.

Data Accessibility section

Open access

You are invited to opt for open access via our author pays publishing model. Payment of open access fees will enable your article to be made freely available via the Royal Society website as soon as it is ready for publication. For more information about open access publishing please visit our website at http://royalsocietypublishing.org/site/authors/open_access.xhtml.

The open access fee is £1,700 per article (plus VAT for authors within the EU). If you wish to opt for open access then please let us know as soon as possible.

Paper charges

Sincerely,
Proceedings B
mailto: proceedingsb@royalsociety.org

Appendix A

Dear Professor Cuthill,

Thank you for allowing us to revise our paper for submission to PRSB. We have addressed all of the referees very useful comments and feel that the paper is much improved. For your convenience, we copy all of the comments below in *italics* followed by our reply and provide a copy of any new text in the manuscript in **bold**.

Addressing these reviews caused our manuscript to exceed the 10-page limit for PRSB, so we have additionally moved several sections to a new supplement to decrease our word count. Details about those changes are at the end of this document in *italics*.

Please let us know if you need any more information

Signed,

Elisa Visher

Review of Visher et al. "The Three Ts of Virulence Evolution During Zoonotic Emergence" for PRSB

General comments:

I found this review well-written and easy to read, the 3-T framework and the introduction compelling, the topic of general interest, and Figures 1 and 3 very interesting/useful. My main concerns, detailed below, are that I thought Table 1 was confusing (in terms of the topic areas and specific studies selected and why) and that the review could benefit from strengthening the connections among the framework, Table 1, and the second half of the review. I also at times had trouble identifying if the traits being discussed were at the scale of the pathogen population or the host population.

Thank you for your kind comments on this manuscript. We are pleased that you found the material engaging and are grateful for your suggestions of important avenues for improvement. We have addressed your concerns point-by-point below.

Detailed comments:

1. I felt there was a disconnect between the introduction/framework, the details in Table 1, and the sections presented later in the review. Strengthening the connections among these would make this review more effective. I recognize the challenges of condensing such a broad literature into a short review. But clarifying in the introductory sections why these specific topics are explored would help focus our attention. Are these a laundry list of questions people have been asking about the current pandemic? Or specific topics that aren't typically discussed, but should be, because they are known to affect the observed dynamics in other systems? Or both?

Thank you for this comment. We have now added a sentence to the introduction, also in response to Reviewer 2 concerns, to specifically say why we focus on these topics. This new line reads :

(Line 74-78): **"Because strong reviews of virulence evolution exist elsewhere in the literature [4,12], our review focuses specifically on virulence evolution in epidemics of novel zoonotic disease to focus on how general theory for virulence evolution is altered by the specific characteristics of emerging zoonotic diseases and shifting selection pressures during epidemics."**

We also feel that changes that we made in response to specific comments by Reviewer 2, (Comment 6) to better transition between the Pareto front and the evolutionary epidemiology theory sections will also help improve these connections.

2. From the introduction, I expected the sections that followed to mirror the 4 questions presented at lines 74-78. Then I was a bit confused at why some of the sections that followed were there since they weren't motivated earlier (e.g., multiple infection, structured populations, environmental transmission, antigenic escape). One way to merge these would be to have main sections that follow the 4 questions noted in 74-78, with subsections within that address critical topics that alter expected outcomes for that main question. I also found section headings framed as questions more compelling than those that were not. Could all the subsections after the framework be framed as specific questions we might/should be asking about pathogen evolution in the SARS-CoV-2 pandemic?

Thank you for these comments, we have edited the 4 questions and the section headings to better parallel each other and clarified the subsections. We have edited the main 4 section heads to be in the form of questions and have edited the subheadings to be more informative.

Line 78-83 now reads: **“Extending beyond the scope of any single theoretical paper on this topic, we will discuss: (1) how do trade-offs between pathogen traits constrain pathogen evolution?; (2) what predicts pathogen virulence at the spillover barrier?; (3) why it is hard to predict how novel zoonotic pathogens will evolve?; and (4) how do optimal strategies in populations with different epidemiological characteristics change over time during an epidemic?”**

And the section headings read:

1. Introduction
2. The Three Ts Framework: Trade-offs, Transmission, and Time Scales
3. How do trade-offs between pathogen traits constrain pathogen evolution?
4. What predicts virulence and transmission rates at spillover?
 - a. Virulence and transmission trade-offs act at spillover
 - b. Virulence and transmission rates of zoonotic pathogens reflect evolutionary histories with their reservoir hosts
5. Why is it hard to predict how a novel zoonotic pathogen will evolve when it spills over into humans?
 - a. Stochastic effects in small populations can overwhelm selection
 - b. Maladapted emerging zoonotic pathogens can evolve in unexpected ways
6. How does a pathogen’s optimal transmission rate and virulence depend on epidemiological characteristics and change over time?
 - a. Selection favors high transmission rates when susceptible density is high at the start of an epidemic
 - b. Structured host populations select for prudent strategies at equilibrium, but transiently select for virulent strategies at the epidemic front, unless there are movement-virulence trade-offs
 - c. How might public health measures shape selection on virulence and transmission rate?
7. Conclusion

Note that we also added a supplement and moved some sections to it because of space constraints. These section headings now read:

- S6(a). Multiple infection can alter selection on virulence depending on the mechanism of competition, and the likelihood of being multiply infected should increase with the force of infection**
- S6(b). Environmental transmission can select for higher pathogen virulence during epidemics and sometimes at equilibrium, unless there are costs associated with environmental persistence**
- S6(c). Antigenic escape can replenish the susceptible host density and persistently select for high transmission rates, unless there are costs associated with antigenic escape**

3. I then expected the topics presented in Table 1 to reflect the sections addressed in the review. This was partially true, but there were additional sections than just those topics presented in Table 1.

We have entirely revised Table 1 so that it now more directly follows the format of the review. For clarity, we have shifted the format so that each row is a Key Finding, rather than an individual paper. There is now a section for each body of theory (or more than one if there is empirical nuance that we present in the body of the paper). Thank you for this suggestion, we agree that this makes the Table much more useful. Note that, along with moving several sections to the supplement, we also moved their associated rows in Table 1 to a new Supplemental Table 1.

New Table 1

Table 1. Empirical tests of virulence evolution theory	
Key Finding	Key Empirical Evidence (Selected Papers)

Virulence and transmission rate are positively correlated through replication rate	Mus musculus / Plasmodium chabaudi [30] ; Homo sapiens / Plasmodium falciparum [31] ; Daphnia magna / Pasteuria ramosa [32] ; Homo sapiens / HIV-1 [33] ; Danaus plexippus / Ophryocystis elektroscirrha [34] ; Meta-analysis of multiple systems [20]
Positive trait correlations saturate so that R_0 peaks at intermediate virulence	Oryctolagus cuniculus / Myxoma virus [17] (virulence-recovery rate); Homo sapiens / Plasmodium falciparum [31] (virulence-transmission rate) ; Daphnia magna / Pasteuria ramosa [32] (virulence rate-transmission rate) ; Homo sapiens / HIV-1 [33] (virulence rate-transmission rate), Danaus plexippus / Ophryocystis elektroscirrha [34] (virulence-transmission rate), Gallus gallus domesticus / Marek's disease virus [35] (virulence- transmission rate), Haemorrhous mexicanu / Mycoplasma gallisepticum [27] (virulence-transmission rate)
High susceptible density at the start of an epidemic selects for higher virulence	Escherichia coli / bacteriophage lambda [36]
Structured host populations select for less transmissible, prudent strategies	Escherichia coli / T4 coliphage [37] ; Plodia interpunctella / granulosis virus [38] ; Escherichia coli / bacteriophage lambda [39]
High virulence can trade-off with decreased host movement	Danaus plexippus / Ophryocystis elektroscirrha [40] ; Haemorrhous mexicanu / Mycoplasma gallisepticum [41] ; Paramecium caudatum / Holospira undulata [42]
Virulence evolves in natural epidemics of emerging disease	Haemorrhous mexicanu / Mycoplasma gallisepticum [43,44] (Less virulent strains spread fastest because of movement-virulence trade-offs and then are replaced by higher virulence strains. When hosts start evolving resistance, virulence continues to increase through increased symptom severity rather than through replication rate) Oryctolagus cuniculus / Myxoma virus [45] (Lower virulence quickly evolves from extremely high virulence introduction strains. When hosts start evolving resistance, virulence starts to increase) Corvus brachyrhynchos / West Nile Virus [46] (A mutation conferring high virulence in American crows was positively selected, though this may have been a result of selection in another bird or vector species)

Table S1: Empirical Tests of Virulence Evolution Theory (Continued)	
Key Finding	Key Empirical Evidence (Selected Papers)
Multiple infection selects for higher virulence	Daphnia magna / Pasteuria ramosa [24] ; Plasmodium chabaudi / Mus musculus [25] ; Panolis flammea / PfNPV [26]
Multiple infection selects for lower virulence	Pseudomonas phaseolicola / Phage $\Phi 6$ / [27] ; Galleria mellonella / Pseudomonas aeruginosa [28] ; Mus musculus / Pseudomonas aeruginosa [29]
Increased environmental transmission selects for higher virulence	Homo sapiens / respiratory tract pathogens [30] ; Tribolium castaneum / Paranosema whitei [31]
Increased environmental persistence can be costly	HeLa cells / vesicular stomatitis virus [15] ; BHK cells / vesicular stomatitis virus [32]
Antigenic escape can be costly	Mus musculus / Influenza A virus [23] ; SARS-CoV-2 yeast system [33]

4. Table 1 could use some explanation about what/why these examples were selected. These examples and topics are a portion of empirical tests of evolutionary theory—why/how were these chosen? It's also not clear how some of these examples fit the trade-off given the strict definition of virulence given in Box 1 (=parasite-induced increase in host mortality rate). E.g., how does the Acevedo review support the trade-off when both virulence and transmission have an increasing relationship with replication rate, and not to each other?

We aimed to provide the key experimental tests of the theory discussed in the evolutionary epidemiology section. To better convey that this table is not comprehensive, we have changed the column title to read “**Key Empirical Evidence (Selected Papers)**”.

Part of the idea for including a virulence definition column was to point out the fact that, while theory tends to use a strict definition of virulence, most biological systems have system-specific deviations from the strict definition of virulence and that moreover virulence must often be measured by proxy because of experimental limitations. While we think that this is an interesting point, we agree that adding it to the table adds too much confusion because we are not able to fully discuss these nuances in the table format. Since we think that we touch on the main points of this issue in the virulence-transmission trade-off section where we talk about how other trade-offs can reproduce similar results, we mostly took virulence definitions out of the table. We have also fully revised the table as above. We keep some discussion of the specific trade-off in the second row where we discuss evidence that R_0 is maximized at intermediate virulence by including the specific trade-off of the system.

To address the specific problem the Acevedo caption, the caption for the Acevedo review was excessively simplified. Virulence and transmission rate show positive relationships to replication rate and to each other, but there is an absence of strong evidence for the saturating relationship between virulence and transmission rate, due to variation in the robustness of experiments.

5. More on table 1 topics: How about empirical studies of virulence or transmission at spillover/introduction vs once established in a population? That seems a topic more relevant to the review, and there are examples from emerging diseases in wildlife (some examples that come to mind include devil facial tumor disease, white-nose syndrome, myxoma virus, west Nile).

Thank you for this idea. We have added a final row to our revised Table 1 (as above) that includes examples where virulence evolves in natural epidemics. We have selected several examples where we feel that there is good evidence of virulence evolution (Myxoma, Mycoplasma, and West Nile) and provided a short sentence about how virulence evolved in response to which selection pressures (when they have been studied). We did check for evidence of virulence evolution in all the examples that Reviewer 1 suggested as well as in several other animal systems, but did not find strong evidence in other systems (Hosts may be evolving resistance to Devil facial tumor, but we did not see any studies suggesting pathogen virulence evolution).

6. Also, text on lines 143 refers to table 1 as evidence of “an increasing number of empirical studies finding support for the core idea” yet not all of these studies support the trade-off hypothesis (as one example, Walther and Ewald), and the idea that support is “increasing” is not evident in the table.

We have now rewritten this sentence. It now reads:

(Line 151-153) **“Given the centrality of the trade-off hypothesis to our understanding of virulence, it is noticeable that there are a number of empirical studies that have found support for the core idea (See Table 1, Rows 1-2) [20].”**

7. 209-231: this section is great, and I learned something new!

Thanks!

8. 237-238. The point that classic models of evolutionary trade-offs between virulence and evolution are evaluated at equilibrium is an important one, and could occur earlier in the paper, either when introducing the 3-Ts or in the trade-off hypothesis section.

We have added a sentence to the “Introduction to the 3Ts” section to introduce this concept. This section now reads:

Lines 109-112 **“This effect further alters a number of theoretical predictions that are classically evaluated at equilibrium for how different host, pathogen, and epidemiological factors shape selection on pathogen traits. Therefore, a pathogen’s optimum strategy changes over time during an epidemic under a wide array of conditions. We will discuss each of these in detail below.”**

9. 241-245: consider replacing “strategies” with “mutants” or “strains”

Thank you for this comment: We have replaced ‘strategies’ with ‘strains’. This section now reads:

Line 277-283: “These models allow for the existence of multiple simultaneous mutants so that the competitive fitness of each can be assessed over shifting epidemiological conditions in time. They show that strains with higher transmission rates and virulence can be selected during epidemic growth stages, despite R_0 optimized (intermediate virulence) strains dominating at endemic equilibrium [19,23]. This is because strains with higher transmission rates spread fastest at the start of the epidemic when the density of susceptible hosts is high [19,23].”

10. L246-253: this paragraph needs clarification. I don't see an intuitive link between mutants with a diversity of transmission rates as discussed in the previous paragraph and the relationships between transmission, generation time, and infectious period early in an epidemic as discussed here. All else being the same (length of generations and infectious periods being equal), a mutant with a higher transmission rate (a higher R_0) will lead to more secondary infections than one with a lower transmission rate (a lower R_0), just based on the larger number of infectious contacts and their subsequent contacts holding a higher proportion of the total infected population as the epidemic grows. I don't see how transmission rate here in any way relates to variation in the length of the infectious period or to generation time. This paragraph needs more explanation to make this connection.

We have rewritten this section to make the central point clear. The key point of virulence-transmission trade-off theory is that infectious period, virulence, transmission rate (beta), and R_0 are all linked to each other so that all else is not the same. We have tried to clarify this by adding the term ‘serial interval’ alongside ‘generation time’ as this term is more commonly used outside of the math literature on this topic and by adding a simplified numerical example to help explain this key point more clearly. This paragraph now reads:

Lines 284-299: “**Intuitively, these results can be explained as: an infected host during the early stages of an epidemic encounters mostly susceptible hosts, so strains with higher transmission rates will have faster growth rates since they have shorter serial intervals (or infection generation times) than strains with higher R_0 (but lower transmission rates) that produce more secondary infections over a longer infectious period but more slowly. For a simplified numeric example, a strain that has an infectious period of 2 days and infects 50% of its 2 contacts per day in an entirely susceptible population will only produce 2 new infections, but will double every 2 days. Comparatively, a strain that has an infectious period of 5 days and infects 40% of its 2 contacts per day in an entirely susceptible population will produce 4 new infections, but only double every 2.5 days. Thus, the higher transmission rate strain can spread faster while susceptible host densities are high during epidemic growth stages, but the R_0 optimized strain can outcompete it when susceptible density is low at endemic equilibrium because it produces a larger number of infections over its longer infectious period. Therefore, improvements in transmission rate are the most important at the start of an epidemic and can be selected for even if they have shorter infectious periods due to increased virulence. This also demonstrates that the high density of susceptible hosts early in epidemics crucially influences selection [12,18,19,23].**”

11. The arguments presented earlier in this section (“selection on virulence and transmission rate during epidemics”), and in Figure 3a’s legend, suggest that early in the epidemic “selection for improved transmission rate can therefore involve decreases, no changes, or increases in virulence depending on the pathogen’s starting point and mutational availability.” Thus, the suggestion on lines 247-250 that in the early stages of the epidemic “strains with higher transmission rates will have faster population growth rates since they have shorter generation times than strains with lower transmission rates...” is confusing to me. Part of my confusion here may be from a lack of clarity in what is meant by generation time (the infection moving from one host to another host? Or a pathogen generation within an infected host?), and what is meant by population growth rate (the replication rate of the pathogen in a host? Or the rate of increase in the number of infected hosts?). But either way I consider the meanings here, I still don't see an intuitive link that increased transmission rate at this stage necessarily links to generation time or duration of infectiousness.

Hopefully the expanded explanation in response to the above comment also addressed this one. On the specific issue of generation time, we have tried to clarify this by adding the term ‘serial interval’ alongside ‘infection generation time’ as this term is more commonly used outside of the math literature on this topic. On the question of how this links to Figure 3a (or Pareto fronts), the editing of Figure 3a to include how selection acts on Pareto front and the expanded introduction to the evolutionary epidemiology theory section in response to Reviewer 2 comments should also help clarify this issue.

12. 255-264: this section is vague. More explanation/examples would help. How does virulence change with multiple infection? What sort of mechanisms for competition or virulence will make what sort of differences in virulence or transmission rate?

We have added extra sentences to this section to explain different pathogen interactions that can lead to selection for lower or higher virulence. This section has also been moved to the supplement. This section now reads:

Line 391-406: **“Multiple infection with pathogens that compete directly for host resources are predicted to be select for higher virulence, while lower virulence can evolve if pathogen outcomes are determined by collective action (cross-reactive immune suppression) or competition for public goods due to less kin selection [2,3]. Because small bottlenecks and short infectious periods in acute respiratory infections do not lead to transmission of diverse infections, high diversity infections may be likely to occur through transmission from multiple infection events [4]. Therefore, the probability of being multiply infected should increase as the number of cases, and thus force of infection, increases since this increases the chance that an individual is infected from multiple sources. Thus, the probability of being multiply infected will vary over the course of the epidemic [5] and any selection effects on virulence due to multiple infection is likely to be weak at the start of epidemic and increase with the number of infected individuals.”**

13. 261-263: Does prevalence need to be high, and the does the epidemic need to be more developed, for multiple infections to occur? Can't diversity be generated within a host's pathogen population during an infection, and then a diverse infection spread?

Previous work with acute, respiratory pathogens and initial evidence from SARS-CoV-2 does not suggest that this is likely. SARS-CoV-2 shows limited with-in host diversity with limited (1-2%) evidence of co-infection [1,2]. Generally, this low diversity is thought to be because of the small bottleneck sizes of transmission (1-8 genomes) and relatively short time window from infection to transmission which mean that, while random mutation generates low frequency variants, these variants do not rise to high frequencies before transmission and so do not transmit though the tight bottleneck [1,3,4]. Because high diversity is not generated within a host, multiple infection can only really occur if an individual is infected from multiple hosts in the same time period or is infected from a host that has an unusually diverse infection (either because they themselves have been co-infected with multiple strains or occasionally if they had an unusually long infection that allowed for longer with-in host evolution [5]). See above for how we have edited this section to address this concern.

14. The section at 255-264 could possibly be combined with the next section on structured host populations? Both deal with population structure (of pathogen or host), and how structure alters evolutionary outcomes. Also, the effect of pathogen diversity is discussed briefly in the host population structure section. Some more information about how pathogen diversity would interact with host structure at different stages of the epidemic would be useful (if known).

We would prefer to keep these sections separate as they address different areas of theory. The multiple infection section deals with co-infection in a single host, while the structured host populations section deals more with exactly who is infecting who (and who that person can then infect). While they are related, as Reviewer 2 does point out, every sub-section in this theory discussion has some overlapping themes (evolution with environmental transmission also depends on host population structure and/or multiple infection and all sections deal with changing selection over time) and we feel that combining them all will decrease reader comprehension. However, we have edited this section head (also in response to Review 1, Major Comment 2) to read (Line 408-409) **“6(c). Structured host populations select for prudent strategies at equilibrium, but transiently select for virulent strategies at the epidemic front”** to specify that we are talking about host population structure in this section. We have also moved the multiple infection section to the supplement.

15. 308-311: the authors briefly touch on selection for antigenic or vaccine escape, and note they will not go into more detail on that topic. If this could be expanded at least a bit, that would be very useful, given that this topic is of great interest to the SARS-CoV-2 pandemic response.

We have added another sentence and more citations to this section. These sentences now read:

Line S47-53: **“We will not fully explore selection for antigenic escape here, but note that selection for antigenic or vaccine escape evolution is significantly slower and less efficient than for drug resistance—likely due to differences in the timing and breadth of with-in host selection pressures [16,18]. Acute infections of respiratory viruses show little with-in host selection for antigenic escape even in vaccinated individuals, likely due to the mismatch in timing between transmission and the activation of the acquired immune system [18–20].”**

16. 333-337: these sentences note how public health measures may alter contacts, spatial structure, symptoms and selection. But they don't go the next step to discuss how those changes could in turn affect virulence or transmission, which are the goals of the review. I understand not wanting to make specific predictions, but for the first point (lines 331-333) you did note how the response might change virulence, and then I was expecting a similar structure to the rest of these, wherein you discussed how each intervention could affect transmission or virulence (and in which direction).

Thank you for these ideas. We have edited these sentences to read:

Lines 338-343: **“Second, decreased travel and extra-household contacts should alter the spatial and social structure of the population to make a more structured transmission network, which might prevent low transmission rate pathogens from spreading initially [63,66]. Third, quarantine of symptomatic individuals may select for decreased or altered symptoms, which could select for lower virulence if symptoms are linked to virulence [71]. Finally, vaccines can sometimes create selection pressures on pathogens with potential evolutionary impacts to consider [72].”**

We felt that we could responsibly follow this structure for points 2 and 3, but we would prefer to avoid making simple statements about vaccination since the evolutionary impacts of vaccination depend so much on the exact mechanism of vaccination and the messaging is fraught. We did cite a Kennedy and Read paper here which we feel responsibly discusses these implications.

17. 370: Edit to “these data are...”

We have edited this. The sentence now reads Line 376: **“These data are exceptionally difficult to quickly gather.”**

Referee: 2

Comments to the Author(s)

1. In this paper, the Visher et al. review existing theoretical literature to further our understanding of the evolution of emerging infectious diseases. There were many things that I liked about this paper. In particular, I found it incredibly easy and enjoyable to read - the authors have a distinctive "voice" that is clear and compelling. I also found Fig. 1 to be very useful. But my major compliment is with the inclusion of the discussion of Pareto fronts - when I started reviewing this paper, I wondered whether I would learn anything new, given existing reviews of virulence evolution and papers studying virulence evolution under non-equilibrium conditions: the Pareto front discussion was novel to me, and thus a very welcome and productive addition to the paper. I do have several comments and suggestions that I think would improve the paper.

Thank you for your kind comments on this manuscript. We are pleased that you found the manuscript enjoyable and are grateful for your suggestions of important avenues for improvement. We have addressed your concerns point-by-point below.

2. As noted above, there have been many primary literature articles and reviews that cover seemingly similar ground to your review (e.g., Bull & Ebert 2008; Bolker et al. 2010 J. R. Soc. Interface 7, 811-822; Cressler et al. 2016; Bonneaud & Longdon 2020). It would be helpful to the reader if there was a clear statement that explains the unique contribution of this paper is by setting it in the context of these seemingly similar papers. For example, you state (lines 71-74), “our aim is to provide a framework so that readers understand the general principles of pathogen virulence and transmission evolution and can also see how variations in the assumptions of these models based upon nuances of biology and population structure can lead to deviations in their predictions.” How is that aim different from the aims of previous papers?

We have added an additional sentence and edited the sentence that you point out here to better explain how our review is different from other reviews and extends beyond single primary literature papers. This section now reads:

Lines 74-83: **“Because strong reviews of virulence evolution exist elsewhere in the literature [4,12], our review focuses specifically on virulence evolution in epidemics of novel zoonotic disease to focus on how general theory for virulence evolution is altered by the specific characteristics of emerging zoonotic diseases and shifting selection pressures during epidemics. Extending beyond the scope of any single theoretical paper on this topic, we will discuss: (1) how do trade-offs between pathogen traits constrain pathogen evolution?; (2) what predicts pathogen virulence at the spillover barrier?; (3) why it is hard to predict how novel zoonotic pathogens will evolve?; and (4) how do optimal strategies in populations with different epidemiological characteristics change over time during an epidemic?”**

3. Another set of major comments concern Fig. 3. I really like the idea of thinking about virulence and transmission evolution via Pareto fronts, as this was not an idea I had encountered before. I actually think you could probably expand on the discussion in this section a bit, as it seemed like there was scope to discuss how the idea of maladaptation might interact with other factors (especially stochasticity and small population sizes) to govern what portion of accessible, sub-optimal phenotype space is actually achievable. That is, it seems like it should be possible to also bound the accessible phenotype space of Fig. 3a from below with biologically infeasible transmission/virulence combinations that are unlikely to survive the highly stochastic early events of spillover (e.g., high virulence, low transmission rate pathogens such as HPAI).

Thank you for this great idea! We have edited Figure 3a to include information about how selection acts on the accessible, sub-optimal region of phenotype space. Essentially, Pareto fronts say which phenotype combinations are at all possible, while selection says which are selected for and against.

New Figure 3

Figure 3a. Conceptual diagram of the Pareto front between virulence and transmission rate. A Pareto front between virulence and transmission rate defines a region of accessible phenotype space. Theory determines where the 'optimal strategy' sits on the Pareto front to determine which regions of this phenotype space are selectively advantaged or disadvantaged. Phenotype combinations far from the Pareto front may technically be possible, but would be highly selectively disadvantaged and likely to go extinct. Possible phenotypes can move towards their optimal strategy along any pathway within the accessible phenotype space. However, we cannot know where a hypothetical phenotype sits below its individual Pareto front. Selection for improved transmission rate can therefore involve decreases, no changes, or increases in virulence depending on the pathogen's starting point and mutational availability.

Figure 3b. Recently emerged viral zoonoses loosely follow a Pareto front of virulence and R_0 where R_0 seems to be maximized at intermediate case fatality rates within viral families. Data is from a published dataset of recently emerged viral zoonoses from mammalian hosts [22]. Approximate R_0 is classified from 1 (no human-to-human transmission) to 4 (endemic transmission). Dots represent potted residuals from linear models of CFR and approximate R_0 including virus family as a factor. By regressing out virus family, we somewhat control for the variation in trade-off shape for each virus and can make general observations across the dataset. Each dot therefore represents the virulence and R_0 of an individual epidemic of viral zoonosis scaled by virus family. Plots were made with 'ggplot2'. See supplement for code and additional figure with data separated by virus family.

4. I also appreciated the attempt to use existing data on human zoonoses in Fig. 3b. However, I feel like there are a lot of unstated assumptions going into Fig. 3b that need to be made explicit, and in so doing may reduce the viability of that figure. In particular, I am uncomfortable with the idea of drawing a single Pareto front for all of these zoonoses. Drawing it this way suggests some kind of universal Pareto front (and thus a universal transmission-virulence relationship) that applies to all emerging infections. But, as you emphasize throughout your paper, the details of this trade-off are likely to be highly system-specific so any universal Pareto front is unlikely to exist.

This is a point that we considered during this analysis and is why Figure 3b is comprised of plotted residuals of models constraining virus family and not the raw data. While we recognize that this is approximate, the idea is that by regressing out virus family, we get a sense of where each virus sits relative to other viruses with similar virulence-transmission relationships. We have edited the figure caption to better capture this and also added an additional figure to the supplement that plots each virus family separately.

See above for edited Figure 3b and caption.

5. However, I wonder if there is a way to examine this idea a bit more quantitatively. As you (ref. 27) and other authors (refs. 53 and 56; also a citation for Farrell and Davies 2018 PNAS 116, 7911 would be appropriate here) have noted, the phylogenetic distance between the reservoir host and the spillover host is an important determinant of transmission, virulence, R_0 , and general "maladaptedness" of pathogens. Thus, I would expect that there should be a relationship between the phylogenetic distance between humans and the reservoir host and the distance to your hypothetical Pareto front. If that is the case, then the far right point on your plot should represent a pathogen that has spilled over from a very close relative, whereas the points near the middle should represent spillovers from more phylogenetically distant hosts. If there is such a relationship, then that would increase my confidence that this universal Pareto front might actually exist, rather than just being drawn on as a visual aid that might not pass biological muster.

This is an interesting way of approaching this figure! We have edited the plot to color the dots by phylogenetic distance. There does not appear to be a strong trend visually (though the previous papers' analyses do show that viruses from more phylogenetically related hosts have higher transmission (R_0 s), and lower case fatality rates). We do agree that this figure mostly serves as a visual aid rather than a robust analysis, which is why we tried to not overstate what this plot says. We have further qualified the title of this plot and it now reads: "**Recently emerged viral zoonoses loosely follow a Pareto front of virulence and R_0 where R_0 seems to be maximized at intermediate case fatality rates within viral families.**"

See above for edited Figure 3b

6. In the section, "Selection on virulence and transmission rate during epidemics" (and really all of the sections that follow this section), it would be helpful to explain how the interpretation of these papers is affected by the notion of

Pareto fronts and no-cost mutations. The difficulty I am having with these sections is that they follow two sections where you basically suggest that trade-offs might not matter very much during spillover (because of stochasticity and no-cost mutations in initially maladapted pathogens), but all of these sections are discussing results that have come out of theoretical papers that have studied virulence evolution in a trade-off context. I found myself having a hard time thinking about when and how evolution might be guided by the predictions of existing theory in these section, given what I had just read about stochasticity and no-cost mutations. Maybe this could be handled by a reordering of the sections? I.e., would it make sense to start the discussion with lines 232-317, and then step back and say, "All of the preceding was focused on evolution in context where stochasticity is unimportant and evolution is constrained by trade-offs. These situations may be quite rare, in general for emerging pathogens." And so on. I'm not sure if that's better, but I thought I'd throw it out there for your consideration.

Thank you for this comment. We agree that the introduction to the theory was abrupt given that we had just spent a lot a time saying how evolution was not predictable. We gave quite a bit of thought to how to introduce this better to keep flow through the manuscript. We decided not to move this section to earlier in the paper as we do think this theory is important and we didn't want to introduce it and then have the discussion of Pareto fronts seem to entirely undermine it. Instead, we've built up more of a discussion throughout the paper to say how selection acts on Pareto fronts, including editing figure 3a to include an idea of how variation on optimal strategy alters how selection on a Pareto front and adding a new paragraph introduction the evolutionary epidemiology theory.

Lines 232-234: **"Therefore, Pareto fronts determine which phenotype combinations are possible, and selection acts upon these possible phenotypes to move them towards more selectively advantageous regions."**

See above for new Figure 3

Lines 246-271: **"6. How does a pathogen's optimal transmission rate and virulence depend on epidemiological characteristics and change over time?"**

While we cannot predict exactly where the virulence and transmission rate of an emerging zoonotic disease sit relative to its Pareto front and thus also cannot predict whether fitness-improving mutations necessarily have costs, evolutionary epidemiology theory can tell us how different epidemiological characteristics shift which regions of the possible phenotype space are selectively advantageous. Additionally, while novel zoonotic pathogens sitting far below their Pareto front may initially have costless fitness-improving mutations, their evolution will be increasingly constrained by trade-offs as their fitness improves and they approach their Pareto front.

Thus, evolutionary epidemiology theory based upon the virulence and transmission trade-off can tell us what scenarios might select for different pathogen virulence and transmission rates. However, evolutionary epidemiology theory on the virulence and transmission trade-off is perhaps more nuanced than commonly appreciated. We've discussed how variations in trade-off shape can lead to different optimal phenotypes for different pathogens [12,17,21], but the optimal values of these rates can also depend on host and parasite epidemiological characteristics and change over time in an epidemic [4,12]. While saturating virulence and transmission rate trade-offs generally predict that intermediate virulence and transmission rate is optimal, certain epidemiological characteristics can bias a system towards selecting for higher transmission rate or less virulence depending on the relative selective importance of either trait. Below, we will discuss several bodies of theory that explore how different epidemiological characteristics effect optimal virulence and transmission rate, specifically focusing on those where the effect of the epidemiological characteristic being explored varies depending on the time scale of the epidemic. There are also several additional sections in the supplement on these effects in systems with multiple infection, environmental transmission ('curse of the pharaoh'), and antigenic escape (Supplemental Materials. S6(a), S6(b), S6(c), and Table S1)."

Minor comments:

Lines 61-63: Can you explain what you mean a bit more here? There have been a number of good reviews of the evolution of virulence that try to capture the nuances of evolution in emerging zoonotic disease in a single place. (This is related to my first comment about what the unique contribution of this paper is to be.)

Addressed as above in response to Reviewer 2, Major Comment 2. This section now reads:

Lines 74-83: **“Because strong reviews of virulence evolution exist elsewhere in the literature [4,12], our review focuses specifically on virulence evolution in epidemics of novel zoonotic disease to focus on how general theory for virulence evolution is altered by the specific characteristics of emerging zoonotic diseases and shifting selection pressures during epidemics. Extending beyond the scope of any single theoretical paper on this topic, we will discuss: (1) how do trade-offs between pathogen traits constrain pathogen evolution?; (2) what predicts pathogen virulence at the spillover barrier?; (3) why it is hard to predict how novel zoonotic pathogens will evolve?; and (4) how do optimal strategies in populations with different epidemiological characteristics change over time during an epidemic?”**

Line 96: Given that you are discussing mathematical models, I think it's a bit confusing to use “in theory” in this sentence, since you are using it to mean “potentially” rather than “in a mathematical model.”

This is a good point. This sentence now reads: (Line 101) **“This maladaptation potentially means that...”**

Box 1: “Together the transmission rate and duration of infectiousness (the inverse of virulence) determine the pathogens R_0 ...” It is inaccurate to say that duration of infectiousness is the inverse of virulence, since duration of infectiousness is also determined by other things (in particular, recovery). Overall, I did not find Box 1 to be particularly useful and think it could be cut.

We agree that Box 1 is not particularly useful and have cut it as doing so will also help free up word count to address other reviewer comments.

Lines 189-190: I know what you mean by “this does not necessarily mean that there will be adaptive evolution,” but it is a bit jarring with the preceding sentence that says that “we expect for there to be selection for improved pathogen fitness,” since “adaptive evolution” IS “selection for improved fitness.” Maybe restate as “this does not necessarily mean that adaptive evolution will be the primary determinant of the evolutionary trajectory of pathogen traits,” or something like that.

Sorry we were not clear here. We edited these sentences to instead read:

Line 198-199 **“Because emerging zoonotic diseases are maladapted to human populations, we certainly expect for selection to favor improved pathogen fitness. However, this does not necessarily mean that pathogens will adaptively evolve [10,13].”**

We felt that this was slightly more concise than what Reviewer 2 suggested and by changing ‘we expect there to be selection’ to ‘we expect for selection to favor’, we remove the implication that selection will necessarily happen.

Finally, addressing these reviews has caused the manuscript to extend over PRSB’s stated 10-page limit. We have attempted to mitigate this by moving several sections to a supplement and cutting a few less important citations. We choose sections to move based upon which were least essential to the narrative. We then also made a supplemental table that is a continuation of our “Table 1: Empirical Tests of Virulence Evolution Theory” in the body of the paper to hold the associated table rows for these sections.

Thus, these sections are now in a supplement:

S6(a). Multiple infection can alter selection on virulence depending on the mechanism of competition, and the likelihood of being multiply infected should increase with the force of infection

S6(b). Environmental transmission can select for higher pathogen virulence during epidemics and sometimes at equilibrium

S6(c). Antigenic escape can replenish the susceptible host density and persistently select for high transmission rates

Table S1: Empirical Tests of Virulence Evolution Theory (Continued)

We reference these sections in the main body of the review. This now reads:

Line 269-271: **“There are also several additional sections in the supplement on these effects in systems with multiple infection, environmental transmission (‘curse of the pharaoh’), and antigenic escape (Supplemental Materials. S6(a), S6(b), S6(c), and Table S1).”**

1. Lythgoe KA *et al.* 2021 SARS-CoV-2 within-host diversity and transmission. *Science* **372**. (doi:10.1126/science.abg0821)
2. Valesano AL, Rumpfelt KE, Dimcheff DE, Blair CN, Fitzsimmons WJ, Petrie JG, Martin ET, Luring AS. 2021 Temporal dynamics of SARS-CoV-2 mutation accumulation within and across infected hosts. *PLOS Pathogens* **17**, e1009499. (doi:10.1371/journal.ppat.1009499)
3. McCrone JT, Woods RJ, Martin ET, Malosh RE, Monto AS, Luring AS. 2018 Stochastic processes constrain the within and between host evolution of influenza virus. *eLife* **7**, e35962. (doi:10.7554/eLife.35962)
4. Morris DH, Petrova VN, Rossine FW, Parker E, Grenfell BT, Neher RA, Levin SA, Russell CA. 2020 Asynchrony between virus diversity and antibody selection limits influenza virus evolution. *eLife* **9**, e62105. (doi:10.7554/eLife.62105)
5. Xue KS, Stevens-Ayers T, Campbell AP, Englund JA, Pergam SA, Boeckh M, Bloom JD. 2017 Parallel evolution of influenza across multiple spatiotemporal scales. *eLife* **6**, e26875. (doi:10.7554/eLife.26875)
6. Debbink K, McCrone JT, Petrie JG, Truscon R, Johnson E, Mantlo EK, Monto AS, Luring AS. 2017 Vaccination has minimal impact on the intrahost diversity of H3N2 influenza viruses. *PLOS Pathogens* **13**, e1006194. (doi:10.1371/journal.ppat.1006194)
7. Cobey S, Larremore DB, Grad YH, Lipsitch M. 2021 Concerns about SARS-CoV-2 evolution should not hold back efforts to expand vaccination. *Nat Rev Immunol* **21**, 330–335. (doi:10.1038/s41577-021-00544-9)

Appendix B

Dear Professor Cuthill,

Thank you for all of your help with this process. We have addressed the reviewer's very useful final comments and feel that these changes much improve the manuscript.

For your convenience, we copy all the comments below in *italics* followed by our reply and provide a copy of any new text in the manuscript in **bold**.

Please let us know if you need any more information.

Signed,

Elisa Visher

Reviewer(s)' Comments to Author:

Referee: 1

Comments to the Author(s)

Review of revision by Visher et al. "The Three Ts of Virulence Evolution During Zoonotic Emergence" for PRSB

The authors did a great job revising the manuscript, and addressed all my comments on the previous version. The revision is a stronger paper, and I can't wait for it to be published, so I can share it with colleagues. I really liked the changes to L261-274 explaining differences in transmission rates and R0.

I have only a few minor suggestions:

Supplemental figure 1 is great, and I hate to have it buried in the SI. Any chance this could be squeezed into the main text? Maybe it could join Figure 3 as an extra panel? (Either way, perhaps have an arrow labeling SARS-CoV-2 on that plot, assuming it's there.)

We are glad that you like this figure! We have moved it into the main text by making a new figure 4 with panel B of Figure 3 and this supplemental figure (leaving panel 3A to stand alone as Figure 3). Because this plot uses data from a dataset published before the pandemic (2019), SARS-CoV-2 is not actually on this graph. We edited the figure caption to reflect that this dataset was published in 2019.

These figures now are:

Figure 3. Conceptual diagram of the Pareto front

between virulence and transmission rate. A Pareto front between virulence and transmission rate defines a region of accessible phenotype space. Theory determines where the 'optimal strategy' sits on the Pareto front to determine which regions of this phenotype space are selectively advantaged or disadvantaged. Phenotype combinations far from the Pareto front may technically be possible but would be highly selectively disadvantaged and likely to go extinct. Possible phenotypes can move towards their optimal strategy along any pathway within the accessible phenotype space. However, we cannot know where a hypothetical phenotype sits below its individual Pareto front. Selection for improved transmission rate can therefore involve decreases, no changes, or increases in virulence depending on the pathogen's starting point and mutational availability.

Figure 4. Recently emerged viral zoonoses loosely follow a Pareto front of virulence and R_0 where R_0 seems to be maximized at intermediate case fatality rates within viral families. Data is from a dataset published in 2019 of recently emerged viral zoonoses from mammalian hosts [22]. Approximate R_0 is classified from 1 (no human-to-human transmission) to 4 (endemic transmission). In figure 4A, dots represent plotted residuals from linear models of CFR and approximate R_0 including virus family as a factor. By regressing out virus family, we somewhat control for the variation in trade-off shape for each virus and can make general observations across the dataset. Each dot therefore represents the virulence and R_0 of an individual epidemic of viral zoonosis scaled by virus family. In figure 4B, CFR and Approximate R_0 are directly plotted and separated by virus family so that the non-aggregated trends could be seen within virus families. In both panels, dots are colored by the phylogenetic distance between humans and the reservoir host. Plots were made with 'ggplot2'. See supplement for code.

Figure 3b legend: replace "potted residuals" with "plotted residuals"

Thank you for catching this. We have revised Figure 3b legend to read "plotted residuals".

L307-309. This is perhaps a splitting of hairs, but I'm not convinced that increased environmental sanitation would select for lower virulence "under the curse of the pharaoh". Sanitation would kill propagules in the environment whether they are long-lived/more virulent or short-lived/less virulent. I can imagine if multi-mode transmission is possible, it could shift the system to more direct transmission, which could support less virulent strains than longer-lived propagules with the option of environmental transmission. But that is more about variation in virulence among transmission modes, not under the umbrella of curse of the pharaoh.

Thank you for this comment. We've now realized that by moving the 'Curse of the Pharaoh' section to the supplement, this statement now is introduced somewhat abruptly and lacks the nuance that the section in the supplement previously provided. The mechanism you mention (multi-mode transmission with direct and environmental routes) is a case sometimes modelled under the 'Curse of the Pharaoh' literature umbrella as in (Day, 2002), but potential biological trade-offs between producing environmentally persistent particles and replication rate complicate this body of theory even further (Boldin & Kisdi, 2012; Ogbunugafor et al., 2013). We have changed this sentence to read "altered pathogen virulence" rather than "lowered pathogen virulence" to reflect the complicated nature of this

theory. To make this section less abruptly introduced and to point towards the nuance in the supplement, we have also re-arranged the order of sentences in this paragraph to lead with those introduced in the main text and pointed towards the appropriate supplement sections for those not introduced in the main text. This section now reads:

However, some of these interventions may also contribute to the selection acting on the pathogen [7,9]. First, decreased travel and extra-household contacts should alter the spatial and social structure of the population to make a more structured transmission network, which might prevent low transmission rate pathogens from spreading initially [63,66]. Second, quarantine of symptomatic individuals may select for decreased or altered symptoms, which could select for lower virulence if symptoms are linked to virulence [71]. Third, increased environmental sanitation decreases environmental transmission, thus potentially selecting for altered pathogen virulence under the ‘curse of the pharaoh’ hypothesis [70] (See Supplementary Material S6(b)). Finally, vaccines can sometimes create selection pressures on pathogens with potential evolutionary impacts to consider [72] (See Supplementary Material S6(c)).

R0 is sometimes written as R0, R0, or R0 (subscript). Standardize.

Thank you for catching this. We have revised the manuscript to consistently read R_0 .